:ᐧ᷉PLOS | ONE

# Performance of patient acuity rating by rapid response team nurses for predicting short-term prognosis

Hyung-Jun Kim[1,2], Hyun-Ju Min[2,3], Dong-Seon Lee[3], Yun-Young Choi[3], Miae Yoon[3], Da-Yun Lee[3], In-ae Song[4], Jun Yeun Cho[2], Jong Sun Park[1,2], Young-Jae Cho[1,2], You-Hwan Jo[5], Ho Il Yoon[1,2], Jae Ho Lee[1,2], Choon-Taek Lee[1,2], Yeon Joo Lee[1,2]*

1 Division of Pulmonary and Critical Care Medicine, Department of Internal Medicine, Seoul National University College of Medicine, Seoul, Republic of Korea, 2 Division of Pulmonary and Critical Care Medicine, Department of Internal Medicine, Seoul National University Bundang Hospital, Seongnam-si, Gyeonggi-do, Republic of Korea, 3 Department of Nursing, Seoul National University Bundang Hospital, Seongnam, Gyeonggi-do, Republic of Korea, 4 Department of Anesthesiology, Seoul National University Bundang Hospital, Seongnam, Gyeonggi-do, Republic of Korea, 5 Department of Emergency Medicine, Seoul National University Bundang Hospital, Seongnam, Gyeonggi-do, Republic of Korea

* yjlee1117@snubh.org

**Data Availability Statement:** All relevant data are within the manuscript and its Supporting Information files.

## Abstract

### Background

Although scoring and machine learning methods have been developed to predict patient deterioration, bedside assessment by nurses should not be overlooked. This study aimed to evaluate the performance of subjective bedside assessment of the patient by the rapid response team (RRT) nurses in predicting short-term patient deterioration.

### Methods

Patients noticed by RRT nurses based on the vital sign instability, abnormal laboratory results, and direct contact via phone between November 1, 2016, and December 12, 2017, were included. Five RRT nurses visited the patients according to their shifts and assessed the possibility of patient deterioration. Patient acuity rating (PAR), a scale of 1–7, was used as the tool of bedside assessment. Other scores, including the modified early warning score, VitalPAC early warning score, standardised early warning score, and cardiac arrest risk triage, were calculated afterwards. The performance of these scores in predicting mortality and/or intensive care unit admission within 1 day was compared by calculating the area under the receiver operating curve.

### Results

A total of 1,426 patients were included in the study, of which 258 (18.1%) died or were admitted to the intensive care unit within 1 day. The area under the receiver operating curve of PAR was 0.87 (95% confidence interval [CI] 0.84–0.89), which was higher than those of modified early warning score (0.66, 95% CI 0.62–0.70), VitalPAC early warning score (0.69, 95% CI 0.66–0.73), standardised early warning score (0.67, 95% CI 0.63–0.70) and cardiac arrest risk triage (0.63, 95% CI 0.59–0.66) (*P*<0.001).

**Funding:** The corresponding author (YJL) was supported by the SNUBH Research Fund (grant no 02-2018-051). The funders had no role in study design, data collection and analysis, decision to publish, or preparation of the manuscript.

**Competing interests:** The authors have declared that no competing interests exist.

## Conclusions

PAR assessed by RRT nurses can be a useful tool for assessing short-term patient prognosis in the RRT setting.

## Introduction

Rapid response team (RRT) is a multidisciplinary team staffed by healthcare professionals with expertise in critical care. Although RRTs provide variable reductions in patient mortality rates, delayed response remains one of the strongest predictors of mortality and unexpected intensive care unit (ICU) admission among critically ill patients [1–3]. Early detection and decisions made by the RRT to correctly triage the patient are essential. The RRT must communicate with medical staff in the general ward, determine whether further medical back-ups are required, and manage the patient during this time.

To facilitate early detection of deteriorating patient status, various early warning scores have been developed. For example, the modified early warning Score (MEWS), one of the most well-known early warning scores, is used in several clinics to facilitate decision-making [2, 4, 5]. However, MEWS provides variable accuracy, and it remains unclear whether the score can be used alone to predict unexpected critical events [3, 6]. Other scoring systems, including VitalPAC early warning score (ViEWS) [7], standardised early warning score (SEWS) [8], and the cardiac arrest risk triage (CART), have also been introduced; however, they were revealed to have performance similar to that of MEWS [3]. These factors have led to the development of several machine learning methods and new risk stratification tools, although the complexity of these techniques can make it difficult to implement these in a pragmatic manner [9, 10].

Patient acuity rating (PAR), a 7-point scale, has been introduced as a method of subjective bedside assessment of the patient [11]. Although it can be a more intuitive method of patient rating, the subjectivity of PAR can be a shortcoming. Nonetheless, recent studies have reported that patients with worse clinical outcomes had higher PAR scores [12, 13], supporting the possibility that PAR can be an effective scoring system. This study aimed to evaluate the performance of the PAR when scored by RRT nurses, versus other early warning scores.

## Materials and methods

### Study design and patients

This study included adult patients who required RRT support during their admission between November 1, 2016, and December 12, 2017. It was performed in accordance with the amended Declaration of Helsinki, and the study protocol was approved by the SNUBH institutional review board (protocol number: B-1604/344-106). The requirement for patient consent was waived due to the impossibility of collecting patient consent during emergency situations.

Sample size was calculated by estimating the area under the receiver operating characteristic curve (AUROC) of PAR as 0.78, according to a pilot analysis of patients from January 2015 to March 2016 in our center, and the AUROC of MEWS as 0.72, according to a previous report [4]. The RRT was activated by the following criteria: systolic blood pressure <90 mmHg; heart rate <50/min or >140/min; respiratory rate <10/min or >30/min; body temperature >39˚C or <36˚C; peripheral oxygen saturation <90% with room air and/or facial mask with oxygen flow >8 L/min; serum pH <7.3; serum partial pressure of carbon dioxide >50 mmHg; serum

partial pressure of oxygen <60 mmHg; serum lactic acid >2.5 mmol/L; serum total carbon dioxide <15 mmol/L; or direct concerns from ward nurses. Patients with sudden cardiac arrest and those in whom PAR was not assessed were excluded from the study. In patients with multiple instances of RRT support, only the first instance was included in the analysis.

Data of baseline demographics, Charlson comorbidity index, department of admission, causes of RRT notification, characteristics of RRT triggering, result of the RRT intervention, and RRT nurse-assessed PAR for the probability of ICU admission and/or mortality within the next day were collected prospectively. All patient data were anonymized before the analysis. This work was supported by from the SNUBH Research Fund (grant no 02-2018-051).

## Patient assessment and score calculation

During the study period, the RRT nurses evaluated the probability of patient deterioration using the PAR, from a scale of 1 to 7 [11]. PAR of 1 corresponds to the lowest probability of patient deterioration and 7 corresponds to the highest. A diagram for better understanding of PAR is available from the original study [11]. Other well-known scores for predicting short-term patient deterioration, including MEWS, ViEWS, SEWS, and the CART score, were calculated afterwards. The scores consisted of the following clinical variables: respiratory rate, heart rate, systolic blood pressure, body temperature and mental status (MEWS); respiratory rate, oxygen saturation, supplementary oxygen use, heart rate, systolic blood pressure, body temperature and mental status (ViEWS); respiratory rate, oxygen saturation, heart rate, systolic blood pressure, body temperature and mental status (SEWS); and respiratory rate, heart rate, diastolic blood pressure and age (CART) [4, 7, 8, 14, 15]. Details of each score calculation are available in the S1–S4 Tables of the supporting information. Patients' vital signs used in score calculation were recorded by ward nurses; these data were immediately sent to the RRT. The RRT nurses and physicians were not aware of these scores at the time of their visits and assessments, and the scores did not influence clinical decisions.

## Nurses in the RRT

Five nurses, who were co-authors of our study, were included in the RRT. All nurses had >9 years of experience of working as a nurse and at least 5 years of experience in the ICU. Nurses 1, 2, and 3 had a >4-year experience of working in the RRT whereas nurses 4 and 5 had a <2-year experience of working in the RRT. Nurses 1, 2, 3, and 4 had completed an ICU nursing program, a nurse preceptor program, and a basic life support provider program. Nurses 1 and 2 had completed an advanced cardiac life support provider program, and nurse 2 was working as an instructor for a basic life support provider program. The RRT nurses did not have direct authority on the final medical decision.

## Study outcomes and statistical analysis

Mortality and/or ICU admission within the next day of RRT activation was considered as the "composite outcome." The patients' scores were used to create receiver operating characteristic curves for predicting the composite outcome. The AUROCs for PAR and other scoring systems were compared using the DeLong method with Bonferroni-adjustment. To assess the goodness-of-fit of a regression model, calibration plot was drawn with Hosmer-Lemeshow test. For analysis of baseline characteristics, categorical variables were reported as number and percentage and were analyzed utilizing the chi-square test or Fisher's exact test. The Mann-Whitney *U* test was performed to describe continuous variables as median and interquartile range (IQR). All statistical analyses and graphing were performed using STATA version 12.0 (StataCorp, College Station, TX, USA). *P*-value of less than 0.05 was considered to indicate

statistical significance. The statistical methods were checked by a qualified statistician from Medical Research Collaborating Center of Seoul National University Bundang Hospital.

## Results

### Patient characteristics

During the study period, 1,441 patients triggered the RRT. Nine patients with sudden cardiac arrest and six patients whom PAR was not assessed were excluded. Therefore, 1,426 patients were included in the final analysis. Among them, 258 patients (18.1%) experienced death and/or ICU admission within 1 day, defined as the "composite outcome".

The patients exhibited male predominance (60.6%), a median age of 72 (IQR, 61–79) years, a mean body mass index of 21.4 (IQR, 19.4–24.7) kg/m$^2$, and a median Charlson comorbidity index of 2 (IQR, 1–4). Patients who died and/or admitted to the ICU seemed to have lower body mass index than those who did not (median 21.2 vs. 22.0, $P = 0.020$). Sex, age, and Charlson comorbidity index did not differ significantly between the two groups ($P = 0.173$, 0.310, and 0.427, respectively). RRT was usually activated in the medical department (76.0%) rather than in other departments. The most common triggering criteria was hypoxemia (38.2%), followed by abnormalities in blood pressure (24.9%), respiratory rate (13.6%), heart rate (10.5%), and partial pressure of carbon dioxide (7.2%). Composite outcome was expected when patients had abnormalities in serum pH (3.9% vs. 0.6%, $P<0.001$) or when the RRT was contacted directly by phone due to other concerns (18.2% vs. 4.5%, $P<0.001$). The median PAR was 3, with significantly higher PAR among those with composite outcome (median, 6; IQR, 4–7) than in the others (median, 3; IQR, 2–4) ($P<0.001$). Other early warning scores were also significantly higher in the composite outcome group (Table 1).

### Comparison of PAR against other scores

The AUROCs for predicting mortality, ICU admission, and the composite outcome within the next day were calculated for PAR and other scores. In predicting the composite outcome, PAR provided an AUROC of 0.87 with a 95% confidence interval (CI) of 0.84–0.89, which was significantly higher compared to MEWS (AUROC, 0.66 [95% CI, 0.62–0.70]; $P<0.001$), ViEWS (AUROC, 0.69 [95% CI, 0.66–0.73]; $P<0.001$), SEWS (AUROC, 0.67 [95% CI, 0.63–0.70]; $P<0.001$) and CART (AUROC, 0.63 [95% CI, 0.59–0.66]; $P<0.001$) (Fig 1). PAR was superior in anticipating ICU admission (AUROC, 0.87 [95% CI, 0.84–0.89]) than MEWS (AUROC, 0.65 [95% CI, 0.61–0.69], $P<0.001$), ViEWS (AUROC, 0.68 [95% CI 0.64–0.72]; $P<0.001$), SEWS (AUROC, 0.66 [95% CI 0.62–0.70]; $P<0.001$), and CART (AUROC, 0.63 [95% CI 0.59–0.67], $P<0.001$). Although MEWS was superior in predicting mortality (AUROC, 0.79 [95% CI 0.70–0.87]) than CART (AUROC, 0.58 [95% CI 0.49–0.67]; $P = 0.001$), it was not superior than MEWS (AUROC, 0.69 [95% CI 0.61–0.76]; $P = 0.062$), ViEWS (AUROC, 0.70 [95% CI 0.61–0.78]; $P = 0.101$), or SEWS (AUROC, 0.69 [95% CI 0.62–0.77]; $P = 0.073$) (Table 2). A cut-off value of PAR $\geq 4$ results in sensitivity of 84.9% and specificity of 73.2% for predicting the composite outcome. The sensitivity and specificity of PAR for various cut-off points are described in Table 3. Calibration plot of PAR for predicting the probability of the composite outcome seemed feasible (Fig 2), but Hosmer-Lemeshow test revealed $P$ of 0.039 with group of 7, and 0.253 with group of 6.

## Discussion

We evaluated the ability of PAR performed by RRT nurses to predict short-term patient prognosis. Compared to other early warning scores, PAR revealed better performance in predicting mortality and/or ICU admission within a day of RRT activation.

**Table 1. Characteristics of patients who were identified using the rapid response team.**

| Variables | Total patients N = 1,426 | Death or ICU admission within 1 day n = 258 | Alive without ICU admission within 1 day n = 1,168 | P |
|---|---|---|---|---|
| Sex | | | | 0.173 |
| Male | 864 (60.6) | 166 (64.3) | 698 (59.8) | |
| Female | 562 (39.4) | 92 (35.7) | 470 (40.2) | |
| Age, years | 72 (61–79) | 73 (63–80) | 71 (60–79) | 0.310 |
| Body mass index, kg/m$^2$ | 21.4 (18.5–24.2) | 22.0 (19.4–24.7) | 21.2 (18.4–24.1) | 0.020 |
| Charlson comorbidity index | 2 (1–4) | 2 (1–4) | 2 (1–4) | 0.427 |
| Department | | | | 0.527 |
| Medical | 1,084 (76.0) | 190 (73.6) | 894 (76.5) | |
| Surgical | 340 (23.8) | 68 (26.4) | 272 (23.3) | |
| Emergency room | 2 (0.1) | 0 (0.0) | 2 (0.2) | |
| Activation criteria | | | | |
| SpO$_2$ | 545 (38.2) | 88 (34.1) | 457 (39.1) | 0.133 |
| Blood pressure | 355 (24.9) | 56 (21.7) | 299 (25.6) | 0.191 |
| Respiratory rate | 194 (13.6) | 40 (15.5) | 154 (13.2) | 0.325 |
| Heart rate | 150 (10.5) | 21 (8.1) | 129 (11.0) | 0.169 |
| PaCO$_2$ | 102 (7.2) | 16 (6.2) | 86 (7.4) | 0.512 |
| Total CO$_2$ | 84 (5.9) | 13 (5.0) | 71 (6.1) | 0.521 |
| PaO$_2$ | 39 (2.7) | 10 (3.9) | 29 (2.5) | 0.214 |
| Body temperature | 29 (2.0) | 7 (2.7) | 22 (1.9) | 0.393 |
| Lactic acid | 22 (1.5) | 5 (1.9) | 17 (1.5) | 0.569 |
| pH | 17 (1.2) | 10 (3.9) | 7 (0.6) | <0.001 |
| Direct call | 100 (7.0) | 47 (18.2) | 53 (4.5) | <0.001 |
| PAR | 3 (2–4) | 6 (4–7) | 3 (2–4) | <0.001 |
| MEWS | 3 (2–5) | 5 (3–6) | 3 (2–5) | <0.001 |
| ViEWS | 8 (6–11) | 10 (8–13) | 8 (5–10) | <0.001 |
| SEWS | 4 (2–5) | 5 (3–7) | 3 (2–5) | <0.001 |
| CART | 16 (9–24) | 21 (12–29) | 13 (9–23) | <0.001 |
| Length of hospital stay | 21 (10–37) | 20.5 (10–39) | 21 (10–36) | 0.963 |
| Overall in-hospital mortality | 353 (24.7) | 113 (43.1) | 240 (20.6) | <0.001 |
| Overall ICU admission | 502 (35.1) | 238 (90.8) | 264 (22.6) | <0.001 |

Values are shown as number (percentage) or median (interquartile range). Abbreviations: SpO$_2$, percent saturation of hemoglobin with oxygen as measured by pulse oximetry; PaCO$_2$, partial pressure of carbon dioxide; PaO$_2$, partial pressure of oxygen; ICU, intensive care unit; PAR, Patient Acuity Rating; MEWS, Modified Early Warning Score; ViEWS, VitalPAC Early Warning Score; SEWS, Standardised Early Warning Score; CART, Cardiac Arrest Risk Triage.

This is the first study to demonstrate the superiority of PAR over other early warning scores in predicting short-term patient prognosis in the RRT setting, highlighting the importance of bedside patient evaluation implemented in previous studies with different clinical settings [16, 17]. PAR does not require complex calculations which makes it easier to utilize in acute bedside settings. The use of a simple assessment such as PAR can accelerate communication between medical professionals, and reduce errors related to score calculation. For example, recorded respiratory rates can be different from the actual rates, and elderly patients may not exhibit abnormal vital signs, despite their critical medical condition [18, 19]. Therefore, there is a risk that the score calculations of other early warning scores are based on incorrectly normal or undetected abnormal vital signs, which might otherwise be detected by a trained healthcare professional.

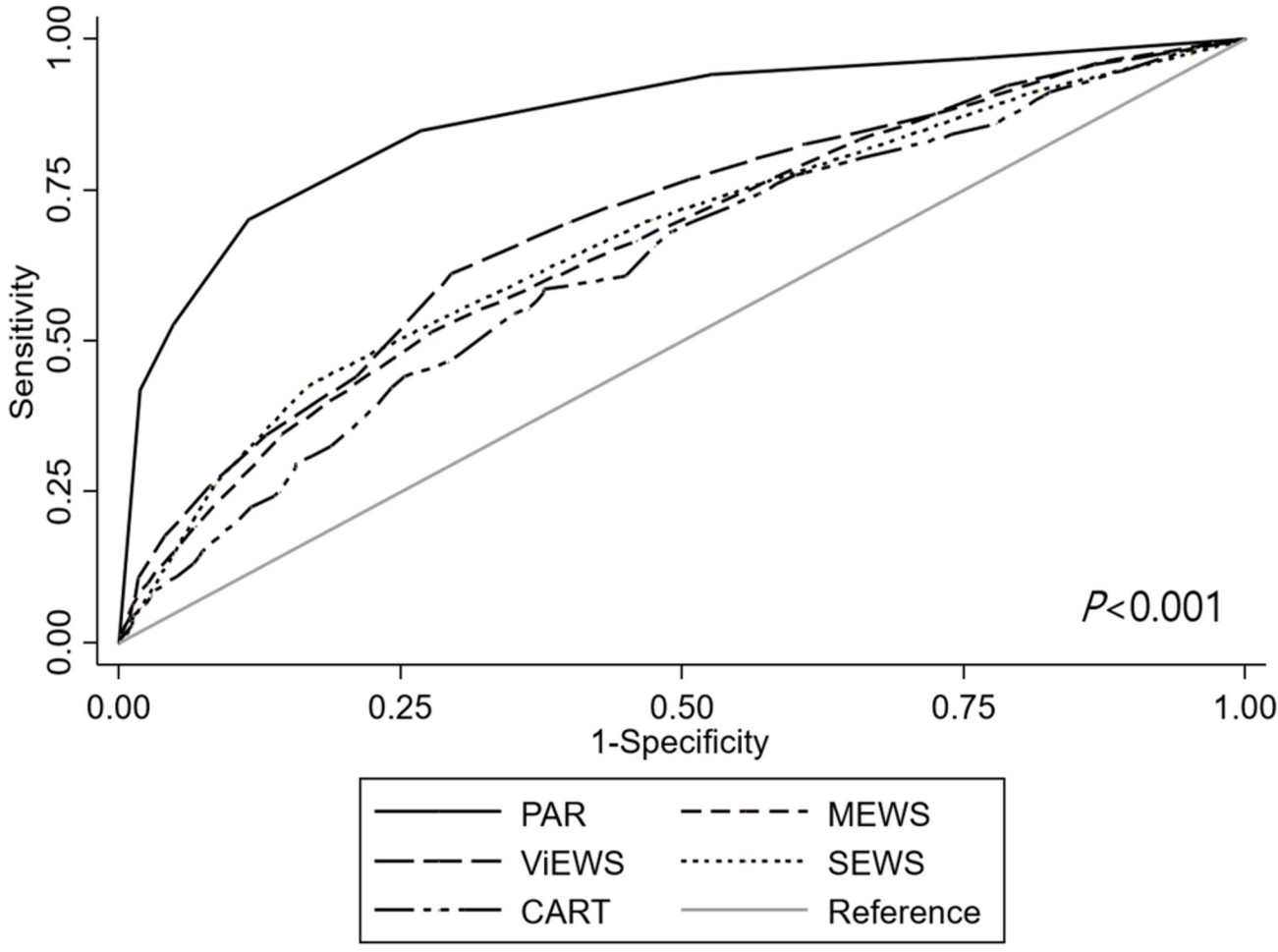

**Fig 1. Receiver operating characteristic curve of each score.** Patient Acuity Rating assessed by the rapid response team nurse was superior in predicting mortality and/or intensive care unit admission within the next day, compared to several other early warning scores.

Our results highlight that non-physician healthcare professionals are helpful in the RRT setting. A previous study has demonstrated the AUROC of PAR assessed by physicians to be 0.82 (0.69 for residents and 0.85 for attendings) for predicting short-term patient deterioration

**Table 2. Areas under the receiver operating characteristic curve of scoring systems for predicting short-term prognoses.**

| Score | Composite outcome [a] | Mortality within 1 day | ICU Admission within 1 day |
|---|---|---|---|
| PAR | 0.87 (0.84–0.89) | 0.79 (0.70–0.87) | 0.87 (0.84–0.89) |
| MEWS | 0.66 (0.62–0.70) | 0.69 (0.61–0.76) | 0.65 (0.61–0.69) |
| ViEWS | 0.69 (0.66–0.73) | 0.70 (0.61–0.78) | 0.68 (0.64–0.72) |
| SEWS | 0.67 (0.63–0.70) | 0.69 (0.62–0.77) | 0.66 (0.62–0.70) |
| CART | 0.63 (0.59–0.66) | 0.58 (0.49–0.67) | 0.63 (0.59–0.67) |

Data are shown as area under the receiver operating characteristic curve (95% confidence interval).

[a] Composite outcome indicates both death and/or ICU admission within 1 calendar day after identification by the rapid response team.

Abbreviations: ICU, intensive care unit; PAR, Patient Acuity Rating; MEWS, Modified Early Warning Score; ViEWS, VitalPAC Early Warning Score; SEWS, Standardised Early Warning Score; CART, Cardiac Arrest Risk Triage.

**Table 3. Sensitivities and specificities of the composite outcome according to the Patient Acuity Rating.**

| PAR | Sensitivity (%) | Specificity (%) |
|---|---|---|
| ≥ 1 | 100.0 | 0.0 |
| ≥ 2 | 96.9 | 24.1 |
| ≥ 3 | 94.2 | 47.4 |
| ≥ 4 | 84.9 | 73.2 |
| ≥ 5 | 70.2 | 88.5 |
| ≥ 6 | 52.7 | 95.2 |
| 7 | 41.9 | 98.1 |

Abbreviation: PAR, Patient Acuity Rating.

[11], which is comparable to the AUROC in this study (0.87 [95% CI 0.84–0.89]). To prevent and reduce physician burnout, placing limitations on duty hours can be effective [20], which may lead to an increased need for non-physician healthcare professionals. In combination with early warning scores, the use of subjective patient assessments by ward nurses can provide a better ability to predict patient deterioration [21]. Considering the sensitivity and specificity of PAR (Table 3), a cut-off value of ≥4 can be reasonable, with good sensitivity and fair specificity for predicting short-term patient deterioration.

The form of RRT in this study (combination of nurse-led rapid response team and a physician-led medical emergency team) has several advantages. First, nurse-driven RRT initiation can lower the risk of afferent limb failure, which is an issue commonly experienced by medical staff in RRTs. Each step during RRT activation is associated with an individual's judgment, and the patient identification process forms a critical part of this activation [22]. If the activation threshold is excessively high, it could lead to afferent limb failure, delayed intervention and patient mortality [23]. Initial nurse-to-nurse contact can be less intimidating to ward nurses than communicating directly with intensivists leading to earlier contact between ward nurses and the RRT. Considering that patients for whom the RRT was contacted directly by phone tended to have worse prognoses in our study, lowering the threshold for RRT activation is necessary for better patient outcomes. Another advantage of our model is its cost-effectiveness in settings with a shortage of exclusive RRT physicians. Although the physician-led medical emergency team approach provides a direct and rapid decision-making process by physicians, it can be difficult to find physicians who are willing to work exclusively in the RRT framework, especially in settings with no financial or legal support for RRT. Aided by the RRT nurses' PAR scores, physicians can quickly understand the status of a patient, leading to efficient management and better outcomes.

Our study identified smaller AUROC values for other early waring scores compared to previous studies [3, 4, 7, 14, 15]. This is likely related to the studies being performed in different medical settings. MEWS was validated in the acute medical admissions setting, and nurses collected the scoring variables during their routine duties [4]. ViEWS and SEWS were studied in the emergency treatment setting [7, 8], and CART was validated in the general ward [15]. Moreover, these tools targeted different clinical outcomes in each studies: admission to a higher dependency unit, attendance of the cardiac arrest team, death and survival at 60 days (MEWS) [4]; in-hospital mortality within 24 hours (ViEWS) [7]; in-hospital mortality and length of stay (SEWS) [8]; and in-hospital cardiac arrest and ICU admission (CART) [15]. These differences could create a noticeable difference in the AUROC values produced, as our study calculated AUROC with selected patients who activated the RRT, in which worse prognosis is anticipated compared to the normal patient population. However, this does not

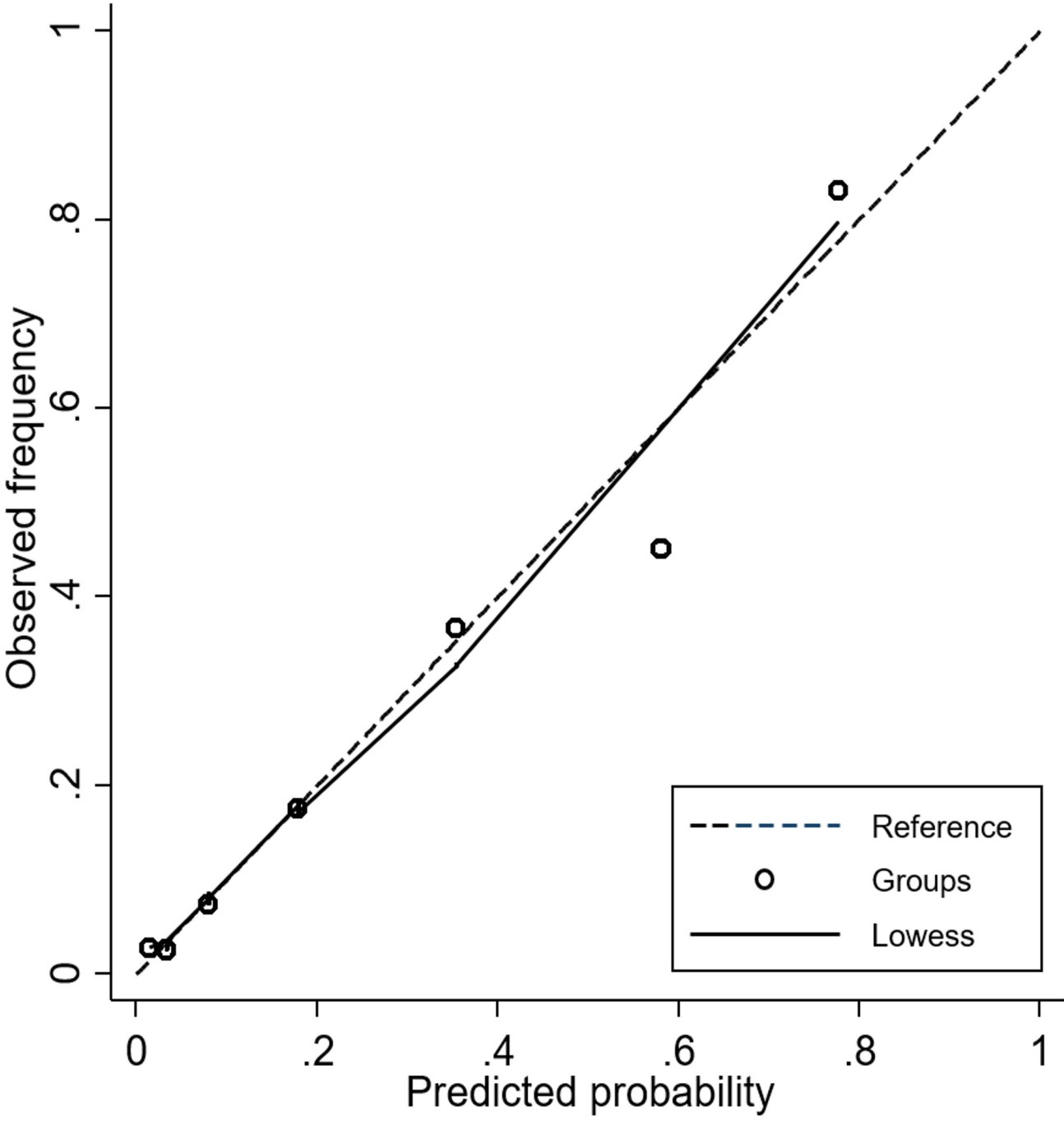

**Fig 2. Calibration plot of Patient Acuity Rating for predicting the probability of the composite outcome.**

devaluate the power of PAR in our study; the AUROC of PAR was 0.87 (95% CI 0.84–0.89), which refers to good power for predicting short-term patient deterioration.

This study has several limitations. First, this study had a single-center design and included only 5 RRT nurses with long working experience, and thus, our results may not be generalizable to other healthcare professionals or less experienced nurses. Inter-observer variability should be considered, and further studies on healthcare professionals with various

backgrounds can enforce the strength of PAR. Second, although final decision of patient management was made by the notified physician, the RRT nurse's subjective opinion could have influenced the physician's decisions, which might have led to more frequent ICU admissions for patients with high PAR. To exclude such potential influences, PAR needs to be assessed by nurses outside the RRT pathway in future studies. Third, this study included patients who have certain extent of systemic illness. This may influence the AUROC of PAR and early warning scores. Fourth, outcomes other than short-term prognosis, including unnecessary ICU admissions, long-term mortality, and functional status, were not included.

## Conclusions

In conclusion, subjective assessment of the patient by an experienced RRT nurse, represented as PAR, reveals good performance in predicting patient prognosis. Although early warning scores may be useful for identifying at-risk patients, direct examinations by healthcare professionals should be emphasized when RRT is activated.

## Supporting information

**S1 Table. Calculation of modified early warning score.**
(DOCX)

**S2 Table. Calculation of VitalPAC early warning score.**
(DOCX)

**S3 Table. Calculation of standardised early warning score.**
(DOCX)

**S4 Table. Calculation of cardiac arrest risk triage.**
(DOCX)

**S1 File. Relevant data.**
(XLSX)

## Acknowledgments

We gratefully acknowledge all the following dedicated Seoul National University Bundang Hospital Medical Alert First Responder (SAFER) team members who participated by sharing their time and experiences with us to improve patients' safety and care: Koung Jin Suh, Hyoung Woo Chang, Joonghee Kim, Dong Jung Kim, Inae Song, Jae-Hyuk Lee, and You Hwan Jo.

We also acknowledge the members of the medical informatics team for preparing the screening system (BESTBOARD) for the SAFER team in Seoul National University Bundang Hospital.

## Author Contributions

**Conceptualization:** Hyung-Jun Kim, Hyun-Ju Min, Yeon Joo Lee.

**Data curation:** Hyun-Ju Min, Dong-Seon Lee, Yun-Young Choi, Miae Yoon, Da-Yun Lee, Inae Song, Yeon Joo Lee.

**Formal analysis:** Hyung-Jun Kim.

**Funding acquisition:** Yeon Joo Lee.

**Investigation:** Hyun-Ju Min, Yun-Young Choi, Miae Yoon, Da-Yun Lee, In-ae Song, Jun Yeun Cho, Jong Sun Park, Young-Jae Cho, You-Hwan Jo, Choon-Taek Lee.

**Methodology:** Hyung-Jun Kim, Hyun-Ju Min, Jun Yeun Cho, Jong Sun Park, Young-Jae Cho, You-Hwan Jo, Ho Il Yoon, Jae Ho Lee, Yeon Joo Lee.

**Project administration:** Hyun-Ju Min.

**Resources:** Hyun-Ju Min, Dong-Seon Lee, Jun Yeun Cho, Jong Sun Park, Young-Jae Cho, You-Hwan Jo, Ho Il Yoon, Jae Ho Lee, Choon-Taek Lee, Yeon Joo Lee.

**Software:** Hyung-Jun Kim.

**Supervision:** Jun Yeun Cho, Jong Sun Park, Young-Jae Cho, You-Hwan Jo, Ho Il Yoon, Jae Ho Lee, Choon-Taek Lee, Yeon Joo Lee.

**Visualization:** Hyung-Jun Kim.

**Writing – original draft:** Hyung-Jun Kim.

**Writing – review & editing:** Hyung-Jun Kim, Hyun-Ju Min, Jun Yeun Cho, Jong Sun Park, Young-Jae Cho, You-Hwan Jo, Ho Il Yoon, Jae Ho Lee, Choon-Taek Lee, Yeon Joo Lee.

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
