## [Decision Letter · Decision Letter 0]

12 Jul 2019

PONE-D-19-17205

Patient Acuity Rating by Nurses in Rapid Response Team

PLOS ONE

Dear Dr. Yeon Joo Lee,

Thank you for submitting your manuscript to PLOS ONE. After careful consideration, we feel that it has merit but does not fully meet PLOS ONE’s publication criteria as it currently stands. Therefore, we invite you to submit a revised version of the manuscript that addresses the points raised during the review process.

ACADEMIC EDITOR: Although it is of interest, the reviewers have raised a number of points which we believe major modifications are necessary to improve the manuscript, taking into account the reviewers' remarks

We would appreciate receiving your revised manuscript by Aug 26 2019 11:59PM. To enhance the reproducibility of your results, we recommend that if applicable you deposit your laboratory protocols in protocols.io, where a protocol can be assigned its own identifier (DOI) such that it can be cited independently in the future. For instructions see: http://journals.plos.org/plosone/s/submission-guidelines#loc-laboratory-protocols

We look forward to receiving your revised manuscript.

Kind regards,

Wisit Cheungpasitporn, MD, FACP

University of Mississippi Medical Center

Academic Editor

PLOS ONE

Journal Requirements:

Reviewers' comments:

Reviewer's Responses to Questions

**Comments to the Author**

1. Is the manuscript technically sound, and do the data support the conclusions?

Reviewer #1: Partly

Reviewer #2: Partly

Reviewer #3: Partly

2. Has the statistical analysis been performed appropriately and rigorously? 

Reviewer #1: No

Reviewer #2: I Don't Know

Reviewer #3: Yes

3. Have the authors made all data underlying the findings in their manuscript fully available?

Reviewer #1: No

Reviewer #2: Yes

Reviewer #3: Yes

4. Is the manuscript presented in an intelligible fashion and written in standard English?

Reviewer #1: No

Reviewer #2: Yes

Reviewer #3: Yes

5. Review Comments to the Author

Reviewer #1: Reviewer's report (1):

REVIEW: Yeon Lee et al.,

General comment: This is an interesting study evaluating a score with good scientific hypothesis which is Patient Acuity Rating by Nurses in Rapid Response Team. The study is important as a clinical tool and would add to the decision making in the subjected group which is people suffering cardiac arrest.

Title:

“Patient Acuity Rating by Nurses in Rapid Response Team”. The title is inappropriate; I think it would be better to address what you intend to do in the study which is score evaluation in the population you made because with this title it looks like a review article e.g evaluation, assessment

1) Abstract: Not acceptable in the current format , the abstract is poorly written, there is no background. I suggest the following to improve the abstract

• Background , the background should not describe the score the authors intend to study, but it should include literatures to support the hypothesis

• Aim of the work is not clear

• Methods : the authors should not include the statement “is that this work is based on a retrospective review of a prospective observational cohort” This make the reader loss interest indeed they should include methodological summary, a short description of the acuity score, if they used additional instruments, I think the study intend to test two scores which should come here, sample volume type of the study should come in the methodology, the study subjects, should include the type of sampling.

• The population of the study is not described whether they are nurse or nurses and physicians, I think this critical , the authors focused on the patient’s outcome however there is a factor related to the observer , so the observer data should be described and included in the statistical analysis

• Results: the results should start with descriptive numbers from the study population followed by the rest, the authors did not use abbreviation for the score also it is mentioned many times, I do not know why they use capital letters in multiple occasions.

• Conclusions: The conclusion should summarize your results in a better attractive way

• I think that abstract needs to be revised by a professional English editor

2) Introduction:

• The introduction suffers a lot of redundancy, it should be more focused.

• The introduction should include the background of the importance of decisions by the rapid response team,

• The score doe not needs to be described in this section but its importance in different population should be highlighted

3) Hypothesis:

• The authors highlighted that they intend to “This study aimed to evaluate the accuracy of the PAR compared to other early warning scores when scored by RRT” the other scoring system needs to be mentioned like A vesrus B

4) Methodology

a. The methodology as the rest of the manuscript suffers a critical lack of focusing issues

b. The authors do not need to mention that this study is based on the others work, they should cite to this score in the introduction and start the methodology by what they did, the work is considered original if it is conducted in another population and or different sample volume.

c. a figure including the score they used is needed for the reader “available at”

J Hosp Med. 2011 October ; 6(8): 475–479. doi:10.1002/jhm.886.

• d. The population of the study is not clearly described whether they are nurse or nurses and physicians, I think this critical , the authors focused on the patient’s outcome however there is a factor related to the observer diversity, so the observer data should be described and included in the statistical analysis. The number also is essential , the demographics within the group is important e.g within the analyzing nurses what is the total years of experience, whether they share the same precision of assessment

e. Ethical Considerations:

Participant identity kept confidential, final report would not contain any identity. Comprehensive explanation for the participants about the questionnaires, the type, purpose of the study and outcome was done, early rejection, or late withdrawal was permissive. Ethical approval was obtained according to the corporate regulation

f. reliability and validity score for the used questionnaires had not been identified this needs to be checked by a statistician, which include a secondary analysis related to the observers

5) Results

a. The results should be revised with new analysis including the observers data as requested

6) Statistical review: the manuscript needs to be seen by a statistician

the study outcome section needs to incorporated into the statistical analysis

7) Discussion:

a. I think the manuscript needs major revision by the authors to decide upon how the discussion will go

b. The discussion is too long and it should start with the salient findings followed by focused analysis

c. The discussion needs 2nd round of revision after fixing the earlier issues

8) Level of interest: An article of importance in its field

9) Quality of written English: Need secondary revision

10) Ethical concerns: The author explained that they obtained waived ethical approval from their institute regarding the patients but they did should include approval related to the participation of nurses

11) References:

a. The references style was not adequately followed according to Plos One style.

12) Declaration of competing interests: I declare that I have no competing interests.

Reviewer #2: OVERALL IMPRESSION

This study is new in that it seeks to assess the accuracy of PAR scores generated by nurses in the context of nurse-initiated Rapid Response Teams. Specifically, it aims to test the ability of the nurse-rated PAR to predict physicians’ decisions regarding ICU admission, and patient mortality, in the following 24 hours. The scientific merit of the study is limited by a lack of information and clarity regarding certain aspects of the methodology.

MAJOR COMMENTS

Materials and Methods

Comment 1: More information is needed on the methods e.g. inclusion/exclusion criteria for both patients and nurses and on what basis the sample size was determined?

Comment 2: Were all the scores independent cases i.e. individual patients?

Comment 3: Line 72: It’s not clear what a “retrospective review of a prospective cohort study” means. What were the reasons for the delay and what was the duration of this between the prospective data collection and the retrospective review? Please clarify and where possible provide a reference for the prospective cohort study on which the retrospective review was based.

Comment 4: Lines 98-101 – were the PAR scores and the four early warning sign scores shared with the intensivists who made the decisions about ICU admission on those occasions when the nurse first raised the alarm?

Comment 5: Line 124: please give more detail on the logistic regression methods i.e. how variables were selected, entered, and removed from, the final models.

Results

Comment 6: Line 169: which statistical test was used to generate this probability level? None of the statistical tests described in the Methods seem appropriate. ANOVA or Kruskal-Wallis would be the tests usually employed for multiple group comparisons such as this.

Discussion

Comment 7: Adding some discussion of the similarities and differences between the authors’ PAR results and those of related studies (e.g. Edelson et al (2011) [ref 10] and O’Donnell et al (2016) [ref 11]) is important in order to place the findings in the context of the available literature - similar to the way the authors discuss their early warning score findings commencing at line 268.

Comment 8: The authors should acknowledge the limitations of their data when discussing the importance of nurse work experience and the accuracy of PAR scores. Specifically, they identified two nurses with less RRT experience whose PAR score accuracy was compared with three more experienced RRT nurses. Such a small sample size (of nurses) cannot provide an adequate test of the impact of experience on PAR accuracy so greater caution in generalising this finding is called for. The least experienced nurses might have had other characteristics which were of equal or greater relevance to the question of their PAR accuracy.

Comment 9: Lines 281-283: the authors rightly identify a crucial limitation of the design i.e. that the nurses involved were not independent assessors of patient acuity but involved in the triage process itself, which would likely inflate the AUROCs of the PAR scores in relation to the admission to ICU outcome. It would be helpful if the authors could reflect at this juncture on improved aspects of design which would provide a more robust test of nurse-scored PAR accuracy in this clinical context.

MINOR COMMENTS

Introduction

Comment 10: Lines 52-59: A slightly more comprehensive summary of the early warning score methods available would strengthen the Introduction – especially so if addressing the pros and cons of all four of the scores used in the study.

Materials and Methods

Comment 11: The authors chose to compare the AUROC of the PAR, a subjective measure, with four objective measures derived from clinical signs. Why did they choose these particular objective measures (see Comment 10)? Did they explore the possibility of making comparisons between PAR and other subjective measures of patient acuity? Explanation of the rationale in both cases would be informative in assessing the validity of the design.

Comment 12: Line 131: “Patterns of distributions were assessed by visual inspection” – what was the rationale for not testing the normality and heterogeneity of variance of the distributions using commonly used tests?

Results

Comment 13: Line 139: “composite outcome” – please clarify the meaning of this phrase at this point.

Comment 14: Line 146: “when the RRT was contacted directly” – please explain what this means – what were the other means by which the RRTs were engaged?

Comment 15: Line 169: Please provide the probabilities associated with the analyses of differences in AUROC’s between the five measures for the ICU admission and mortality outcomes.

Comment 16: Lines 193-196 and Lines 200-202: Please quote the probabilities associated with these three analyses.

Comment 17: The analyses by individual nurses, taking into consideration their levels of experience, and the logistic regression modelling of the predictive value of PAR and ViEWS in combination bring added value to the design and are of practical significance.

Discussion

Comment 18: Are the sentences at lines 215 and 223 not contradictory?

Reviewer #3: Thank you for allowing me to review this study by Kim et al. Rapid response teams are a major development in inpatient care and understanding better ways to objectively triage and assess patients is an important issue. I also think the more we can understand how all providers of care (nurses, physical therapists, physicians) interact with patients is extremely useful. Thus, I read this study with great interest.

The authors used a retrospective review of a prospective observational cohort of patients at a single health center and how patients were assessed by rapid response team nurses. They reported finding that a subjective, 7-point Likert scale, the PAR performed better overall than other, more objective scores. The authors also spent a lot of time dissecting the experience of one less-experienced team member versus the other four.

I think that there is some interest in the overall finding - PAR performed better than other scores. I am very guarded in my review of the study however because the authors used almost half of the manuscript to discuss essentially a descriptive difference between 2 nurses newer to the team to 3 others who had been on the team for a longer period of time. This study was not designed to assess the experience of the nurses on a rapid response team (statistics cannot be performed this low of a sample size). I thus think this study presentation needs to be rebuilt around the actual findings of the study.

Major issues:

1) There is no power to assess for the experience of nurses and the authors report experience of nurses as a major conclusion of the entire study

There are 3 nurses with >4 years experience as an RRT nurse. There are 2 with 1 year or less experience. Only 1 of 2 nurses with less experience (Nurse 4) had a significantly lower AUROC than the 3 more experienced nurses (Nurses 1-3). There is no way to make even a subjective conclusion off a single individual underperforming (Nurse 4) and the other individual performing the same (Nurse 5). I don’t even think it belongs in the manuscript beyond perhaps a description of the 5 nurses in the RRT team. I would draw the conclusion the nurses performed the same except for one. They also had less training, but another nurse with even less training (Nurse 5) performed the same.

Because this is reported as a major conclusion of the study throughout, including the abstract, this entire manuscript needs to be refocused on the actual objective study findings before it can be fully vetted.

2) I am not sure the comparison scores are the best available

The authors admit that MEWS was not developed for general ward patients (it was developed for acute medical admissions). The scores also looked at different outcomes. Why did the authors choose these scores? Is it fair to say the PAR score outperforms scores that are not clearly validated in the actual patient population? Is this important? ViEWS and CART seem fair, but SEWS, and most certainly MEWS, seem less applicable. Are there better scores to compare? The authors calculate the Charlton Comorbidity Index as important but did not compare the PAR with this score. I think it would be helpful.

3) What was the best cut-off to predict ICU admission for the PAR in this study?

I was a bit confused that the PAR score is deemed more reliable a determination for predicting bad outcomes in patients who have an RRT consultation, however, it was not expressly clear to me what cut-off was used to predict bad outcome most accurately. Certainly, if one is to apply the use of PAR for clinical practice, there should be some cutoff point for providers to follow as to should the patient be admitted to the ICU. The authors need to discuss this as it is critical for actual applying a diagnostic test.

4) One combination assessment was tested – PAR + ViEWS – but not others

It was reported in Table 4 about the individual nurse subjects’ performance on PAR, ViEWS and PAR+ViEWS. As stated above, the comparison between nurses for experience is a minor point. It would be much more useful to assess if combining the PAR with the ViEWS in the overall cohort was superior to either alone.

5) Authors draw conclusions about nurses in general, but only tested five RRT nurses and did not compare performance to other providers in the same cohort of patients

The authors draw many conclusions on nurses overall assessments versus the performance of other providers (physicians, for example), however this is a highly trained and experienced subset of nurses. There was also no comparison between different types of providers (advanced practice providers, interns, residents, attending physicians) in this particular cohort of patients. To truly make a comparison of these types of providers ability to assess the PAR accurately, their performance on the PAR would need to examined in the same cohort of patients. I think minimal conclusions in direct comparison can be drawn from this study. I also think generalizing the performance of RRT nurses to all nurses throughout the hospital, with a wide subset of skills and specialties, is impossible.

6) Were any further outcomes assessed? If ICU admission at a later time is an outcome, what was the overall outcome of the patients?

Not all ICU admissions are appropriate and more evidence is coming out that perhaps needless ICU admissions are without harm. I also am interested that mortality was actually not significantly predicted by the PAR, which might suggest the ICU admissions did not correlate with mortality as much. Was there any more data to be gained? I would be very interested in overall in-patient mortality. Additionally, if available, 6 month and 1 year mortality would be very useful, although I realize this may be difficult to obtain.

Minor issues:

1) In table 1, need to put sex, female statistics (how many total, death/ICU admission, alive without ICU admission within 1 day)

2) What was the p value comparing in sex, age, BMI Charlson comorbidity index in Table 1? Was it if that variable predicted outcome on logistic regression? State this in the table details.

3) Table 1, don’t need to state detailed criteria for rapid response team in the caption

4) The details of how the RRT work at the hospital is long and extremely detailed. I’m not sure we need all the shift times.

5) What is a part-time intensivist? Is this a critical care attending physician? Is it a moon-lighter in another specialty? The term is imprecise and I do not know what it means. Also, there was mention of ICU fellows, but in what subspecialty? Critical care medicine?

6) When were clinical variables for the various scores obtained? At bedside or last vital sign check? Same day labs or admission labs?

7) Imprecise language in discussion paragraph 3 (page 12). “the possibility that non-physician healthcare professionals can be helpful in the RRT setting”. Certainly no one would argue that a non-physician healthcare professional is not helpful in an RRT. I don’t understand what the point of the statement is but it could be construed as hurtful to imply the care team is not helpful.

8) Discussion paragraph 3, page 12: ACGME duty hours is not referenced. I thought that the ACGME went back to allowing 28 hour shifts for interns. There needs to be an accurate depiction of what the policy is with up to date citations.

6. PLOS authors have the option to publish the peer review history of their article (what does this mean?). If published, this will include your full peer review and any attached files.

Reviewer #1: Yes: Dr Amr Salah Omar, MD, PhD

Reviewer #2: No

Reviewer #3: No

---

## [Author Response · Author response to Decision Letter 0]

28 Aug 2019

Dear editor: 

We would like to thank all the editors and reviewers for providing us with their valuable opinions about our manuscript. We have revised our manuscript according to the comments and recommendations of the reviewers. We have highlighted all changes in the revised manuscript in red font, and a clean copy of the manuscript is also uploaded. Below, we have included an itemized series of responses to the comments of the reviewers.

Reviewer #1: 

1. Title: “Patient Acuity Rating by Nurses in Rapid Response Team”. The title is inappropriate; I think it would be better to address what you intend to do in the study which is score evaluation in the population you made because with this title it looks like a review article e.g evaluation, assessment

Response) We appreciate your comment. After careful discussion between the authors, we have decided to change to title of the article as: “Performance of Patient Acuity Rating by Rapid Response Team Nurses for Predicting Short Term Patient Prognoses.” 

2. Abstract: Not acceptable in the current format, the abstract is poorly written, there is no background. I suggest the following to improve the abstract. The background should not describe the score the authors intend to study, but it should include literatures to support the hypothesis. Aim of the work is not clear

Response) We agree with your comment. We have deleted unnecessary descriptions, and have clarified the aim of our study.

“Background: Although scoring and machine learning methods have been developed to predict patient deterioration, bedside assessment by nurses should not be overlooked. This study aimed to evaluate the performance of subjective bedside assessment of the patient by the rapid response team (RRT) nurses in predicting short-term patient deterioration.”

3. Methods : the authors should not include the statement “is that this work is based on a retrospective review of a prospective observational cohort” This make the reader loss interest indeed they should include methodological summary, a short description of the acuity score, if they used additional instruments, I think the study intend to test two scores which should come here, sample volume type of the study should come in the methodology, the study subjects, should include the type of sampling.

Response) Thank you for your comment. We have deleted the statement “this work is based on a retrospective review of a prospective observational cohort,” and added a short description of the study subjects, along with patient acuity score and other early warning scores.

“Methods: Patients noticed by RRT nurses based on the vital sign instability, abnormal laboratory results, and direct contact via phone between November 1, 2016, and December 12, 2017, were included. Five RRT nurses visited the patients according to their shifts and assessed the possibility of patient deterioration. Patient acuity rating (PAR), a scale of 1–7, was used as the tool of bedside assessment. Other scores, including the modified early warning score, VitalPAC early warning score, standardised early warning score, and cardiac arrest risk triage, were calculated afterwards. The performance of these scores in predicting mortality and/or intensive care unit admission within 1 day was compared by calculating the area under the receiver operating curve.”

4. The population of the study is not described whether they are nurse or nurses and physicians, I think this critical, the authors focused on the patient’s outcome however there is a factor related to the observer, so the observer data should be described and included in the statistical analysis

Response) The observers in our study are nurses from the RRT, and we have clarified our sentences.

“Methods: … Five RRT nurses visited the patients according to their shifts and assessed the possibility of patient deterioration. Patient acuity rating (PAR), a scale of 1–7, was used as the tool of bedside assessment. …”

5. Results: the results should start with descriptive numbers from the study population followed by the rest, the authors did not use abbreviation for the score also it is mentioned many times, I do not know why they use capital letters in multiple occasions.

Response) The numbers from the study population, along with those with main outcomes (composite outcome, defined as death or admission to the ICU within the next day of RRT activation) were added. The scores were abbreviated as they were mentioned in multiple occasions.

“Results: A total of 1,426 patients were included in the study, of which 258 (18.1%) died or were admitted to the intensive care unit within 1 day. The area under the receiver operating curve of PAR was 0.87 (95% confidence interval [CI] 0.84–0.89), which was higher than those of modified early warning score (0.66, 95% CI 0.62–0.70), VitalPAC early warning score (0.69, 95% CI 0.66–0.73), standardised early warning score (0.67, 95% CI 0.63–0.70) and cardiac arrest risk triage (0.63, 95% CI 0.59–0.66) (P<.001). The area under the receiver operating curve of PAR tended to be larger for nurses with longer experience in the RRT.”

6. Conclusions: The conclusion should summarize your results in a better attractive way

Response) We agree that unnecessary sentences were included in the conclusion. It was summarized as following:

“Conclusions: PAR assessed by RRT nurses can be a useful tool for assessing short-term patient prognosis in the RRT setting.”

7. I think that abstract needs to be revised by a professional English editor

Response) We have revised our manuscript with help of a professional English editor website (https://www.editage.co.kr/).

8. The introduction suffers a lot of redundancy, it should be more focused. The introduction should include the background of the importance of decisions by the rapid response team. The score does not needs to be described in this section but its importance in different population should be highlighted

Response) We agree with you about the redundant expressions in the manuscript. Unnecessary sentences were deleted, and we have emphasized the importance of early detection and management of critically ill patients by the RRT. Detailed description about patient acuity rating was moved to the methods section.

“Introduction

Rapid response team (RRT) is a multidisciplinary team staffed by healthcare professionals with expertise in critical care. Although RRTs provide variable reductions in patient mortality rates, delayed response remains one of the strongest predictors of mortality and unexpected intensive care unit (ICU) admission among critically ill patients [1-3]. Early detection and decisions made by the RRT to correctly triage the patient are essential. The RRT must communicate with medical staff in the general ward, determine whether further medical back-ups are required, and manage the patient during this time.

To facilitate early detection of deteriorating patient status, various early warning scores have been developed. For example, the modified early warning Score (MEWS), one of the most well-known early warning scores, is used in several clinics to facilitate decision-making [2, 4, 5]. However, MEWS provides variable accuracy, and it remains unclear whether the score can be used alone to predict unexpected critical events [3, 6]. Other scoring systems, including VitalPAC early warning score (ViEWS) [7], standardised early warning score (SEWS) [8], and the cardiac arrest risk triage (CART), have also been introduced; however, they were revealed to have performance similar to that of MEWS [3]. These factors have led to the development of several machine learning methods and new risk stratification tools, although the complexity of these techniques can make it difficult to implement these in a pragmatic manner [9, 10].

Patient acuity rating (PAR), a 7-point scale, has been introduced as a method of subjective bedside assessment of the patient [11]…”

9. The authors highlighted that they intend to “This study aimed to evaluate the accuracy of the PAR compared to other early warning scores when scored by RRT” the other scoring system needs to be mentioned like A versus B

Response) We understand your concern. We aimed to compare multiple scores against patient acuity rating: modified early warning score, VitalPAC early warning score, standardised early warning score, and cardiac arrest risk triage. We have amended the sentence as following:

“This study aimed to evaluate the performance of the PAR when scored by RRT nurses, versus other early warning scores”

10. The methodology as the rest of the manuscript suffers a critical lack of focusing issues. The authors do not need to mention that this study is based on the others work, they should cite to this score in the introduction and start the methodology by what they did, the work is considered original if it is conducted in another population and or different sample volume.

Response) We appreciate your careful reading, but our study was not based on others’ work: it was from our center. We have deleted the statement “this work is based on a retrospective review of a prospective observational cohort,” and amended the sentences to minimize the misunderstanding by the readers. 

“Materials and Methods

Study design and patients

This study included adult patients who required RRT support during their admission between November 1, 2016, and December 12, 2017.”

11. a figure including the score they used is needed for the reader “available at” J Hosp Med. 2011 October; 6(8): 475–479. doi:10.1002/jhm.886.

Response) We agree with your comment. However, due to the copyright issues, the figure itself could not be used in our manuscript. Therefore, we added a sentence in the methods section as following:

“... PAR of 1 corresponds to the lowest probability of patient deterioration and 7 corresponds to the highest. A diagram for better understanding of PAR is available from the original study [11].…”

12. The population of the study is not clearly described whether they are nurse or nurses and physicians, I think this critical, the authors focused on the patient’s outcome however there is a factor related to the observer diversity, so the observer data should be described and included in the statistical analysis. The number also is essential, the demographics within the group is important e.g within the analyzing nurses what is the total years of experience, whether they share the same precision of assessment

Response) We apologize for the ambiguous description. Nurses were the observers who checked patient acuity rating among critically ill patients contacted to the RRT. Physicians did not participate in the scoring process. We have changed the subtitle in the methods section from “Composition of the RRT” to “Nurses in the RRT.” Detailed information about each nurse are described as following in the manuscript:

“Nurses in the RRT

Five nurses, who were co-authors of our study, were included in the RRT. All nurses had >9 years of experience of working as a nurse and at least 5 years of experience in the ICU. Nurses 1, 2, and 3 had a >4-year experience of working in the RRT whereas nurses 4 and 5 had a <2-year experience of working in the RRT. Nurses 1, 2, 3, and 4 had completed an ICU nursing program, a nurse preceptor program, and a basic life support provider program. Nurses 1 and 2 had completed an advanced cardiac life support provider program, and nurse 2 was working as an instructor for a basic life support provider program. The RRT nurses did not have direct authority on the final medical decision.”

13. reliability and validity score for the used questionnaires had not been identified this needs to be checked by a statistician, which include a secondary analysis related to the observers

Response) Thank you for your comment. The article was checked by a qualified statistician from Medical Research Collaborating Center of Seoul National University Bundang Hospital.

To provide the reliability of PAR in predicting the composite outcome, we drew a calibration plot with Hosmer-Lemeshow test for goodness-of-fit. Calibration plot seemed feasible, but Hosmer-Lemeshow test revealed p-value of 0.039 with group of 7, and 0.253 with group of 6. The calibration plot was included as Fig 2, and the main text was revised as following:

“Methods

Study outcomes and statistical analysis

…The AUROCs for PAR and other scoring systems were compared using the DeLong method with Bonferroni-adjustment. Because PAR involves subjective assessment of patient status, the AUROCs of PAR for predicting the composite outcome according to each participating nurse were analyzed. A combined model of PAR with ViEWS, which showed the largest AUROC among the calculated scores, was built with logistic regression analysis, and the AUROC of this model was calculated as well. To assess the goodness-of-fit of a regression model, calibration plot was drawn with Hosmer-Lemeshow test.

…

The statistical methods were checked by a qualified statistician from Medical Research Collaborating Center of Seoul National University Bundang Hospital.

Results

Comparison of PAR against other scores

 … A cut-off value of PAR ≥4 results in sensitivity of 84.9% and specificity of 73.2% for predicting the composite outcome. The sensitivity and specificity of PAR for various cut-off points are described in Table 3. Calibration plot of PAR for predicting the probability of the composite outcome seemed feasible (Fig 2), but Hosmer-Lemeshow test revealed P of 0.039 with group of 7, and 0.253 with group of 6.

Fig 2. Calibration plot of Patient Acuity Rating for predicting the probability of the composite outcome”

Validity (performance) of PAR in predicting the composite outcome was the main outcome our study tried to assess, by comparing the area under the receiver operating characteristic curve versus other scoring systems. PAR was introduced as a subjective method of assessment of patient deterioration from a previous study (Edelson DP et al. J Hosp Med, 2011), therefore our population for validation is different from the originally derived population. The validity of other early warning scores has been introduced as area under the receiver operating characteristic curve in a previous review (Churpek MM et al. Chest, 2013).

14. The results should be revised with new analysis including the observer’s data as requested

Response) Thank you for the comment. The results according to each observer (nurse) is described in Table 4, and the results section as following: 

“PAR assessment according to each nurse

 The AUROCs were larger for nurses 1, 2 and 3 with longer experience in the RRT (AUROC, 0.91 95% CI 0.87–0.95], 0.91 [95% CI 0.85–0.96] and 0.90 [95% CI 0.86–0.94] respectively) than for nurses 4 and 5, with shorter experience (AUROC, 0.78 [95% CI 0.72–0.84] and 0.80 [95% CI 0.71–0.90] respectively). AUROC of PAR for nurse 4 was significantly smaller compared to those of nurses 1, 2 and 3. 

To compensate for the smaller AUROC of less-experienced nurses’ PAR, we utilized a logistic regression model of PAR with the warning score of the largest AUROC in our study: ViEWS. The combined model of PAR and ViEWS in the overall population showed a significantly improved AUROC (0.875 [95% CI, 0.849–0.900]) compared to that of PAR alone (0.868 [95% CI 0.843–0.894]). This change was mainly due to the improvement in AUROC for nurse 4; the combined model showed a significantly improved AUROC (0.81 [95% CI 0.75–0.86]) for predicting patient outcome compared to that of PAR alone (0.78 [95% CI 0.72–0.84]). Meanwhile, significant improvement of AUROC in this model was not observed for the other four nurses (Table 4).”

15. Statistical review: the manuscript needs to be seen by a statistician. The study outcome section needs to incorporated into the statistical analysis

Response) We appreciate your comment. The article was checked by a qualified statistician from Medical Research Collaborating Center of Seoul National University Bundang Hospital. Fig 2 was added to further assess the reliability of Patient Acuity Rating in predicting the composite outcome, and the term “performance” was used rather than “accuracy.” DeLong method with Bonferroni-adjustment for comparing the area under the receiver operating characteristic curve was double checked by the statistician. A sentence was added to the methods section as following:

“The statistical methods were checked by a qualified statistician from Medical Research Collaborating Center of Seoul National University Bundang Hospital.”

16. I think the manuscript needs major revision by the authors to decide upon how the discussion will go. The discussion is too long and it should start with the salient findings followed by focused analysis. The discussion needs 2nd round of revision after fixing the earlier issues

Response) We apologize for the redundant expressions. The discussion section was shortened, only leaving necessary sentences.

17. Quality of written English: Need secondary revision

Response) We have revised our manuscript with help of a professional English editor website (https://www.editage.co.kr/).

18. Ethical concerns: The author explained that they obtained waived ethical approval from their institute regarding the patients but they did should include approval related to the participation of nurses

Response) We appreciate your comment. The five nurses are co-authors of our manuscript (Hyun-Ju Min, Dong-Seon Lee, Yun-Young Choi, Miae Yoon, and Da-Yun Lee), who all agreed to the design of the study and the final version of the manuscript. It was added to the methods section as following:

“Nurses in the RRT

Five nurses, who were co-authors of our study, were included in the RRT.”

19. The references style was not adequately followed according to Plos One style.

Response) We have double checked our references to be concordant with Plos One style.

 

Reviewer #2: 

MAJOR COMMENTS

1. More information is needed on the methods e.g. inclusion/exclusion criteria for both patients and nurses and on what basis the sample size was determined?

Response) We appreciate your comment. The inclusion/exclusion criteria, along with sample size calculation was included in the manuscript. Sample size was calculated by estimating AUROC of PAR as 0.78 according to a pilot analysis of patients from January 2015 to March 2016 in our center, and AUROC of MEWS as 0.72 according to a previous report (Subbe CP et al. Qjm. 2001;94(10):521-6). With alpha and beta set as 0.05 and 0.20 respectively, the calculated sample size was 1420. 

“Materials and Methods

Study design and patients

This study included adult patients who required RRT support during their admission between November 1, 2016, and December 12, 2017. ...

Sample size was calculated by estimating the area under the receiver operating characteristic curve (AUROC) of PAR as 0.78, according to a pilot analysis of patients from January 2015 to March 2016 in our center, and the AUROC of MEWS as 0.72, according to a previous report [4]. The RRT was activated by the following criteria: systolic blood pressure <90 mmHg; heart rate <50/min or >140/min; respiratory rate <10/min or >30/min; body temperature >39 °C or <36 °C; peripheral oxygen saturation <90% with room air and/or facial mask with oxygen flow >8 L/min; serum pH <7.3; serum partial pressure of carbon dioxide >50 mmHg; serum partial pressure of oxygen <60 mmHg; serum lactic acid >2.5 mmol/L; serum total carbon dioxide <15 mmol/L; or direct concerns from ward nurses. Patients with sudden cardiac arrest and those in whom PAR was not assessed were excluded from the study. In patients with multiple instances of RRT support, only the first instance was included in the analysis.”

2. Were all the scores independent cases i.e. individual patients?

Response) We apologize for the ambiguous description. All the scores were individual patients. If one patient was notified to the RRT multiple times, only the first case was included in the study. The clarified sentence was added to the methods section:

“… Patients with sudden cardiac arrest and those in whom PAR was not assessed were excluded from the study. In patients with multiple instances of RRT support, only the first instance was included in the analysis. …”

3. Line 72: It’s not clear what a “retrospective review of a prospective cohort study” means. What were the reasons for the delay and what was the duration of this between the prospective data collection and the retrospective review? Please clarify and where possible provide a reference for the prospective cohort study on which the retrospective review was based.

Response) Thank you for your comment. This study was not based on others’ work: it was from our center. We have deleted the statement “this work is based on a retrospective review of a prospective observational cohort,” and amended the sentences to minimize the misunderstanding by the readers. 

“Materials and Methods

Study design and patients

This study included adult patients who required RRT support during their admission between November 1, 2016, and December 12, 2017. ...

Sample size was calculated by estimating the area under the receiver operating characteristic curve (AUROC) of PAR as 0.78, according to a pilot analysis of patients from January 2015 to March 2016 in our center, and the AUROC of MEWS as 0.72, according to a previous report [4]. The RRT was activated by the following criteria: systolic blood pressure <90 mmHg; heart rate <50/min or >140/min; respiratory rate <10/min or >30/min; body temperature >39 °C or <36 °C; peripheral oxygen saturation <90% with room air and/or facial mask with oxygen flow >8 L/min; serum pH <7.3; serum partial pressure of carbon dioxide >50 mmHg; serum partial pressure of oxygen <60 mmHg; serum lactic acid >2.5 mmol/L; serum total carbon dioxide <15 mmol/L; or direct concerns from ward nurses. Patients with sudden cardiac arrest and those in whom PAR was not assessed were excluded from the study. In patients with multiple instances of RRT support, only the first instance was included in the analysis.”

4. Lines 98-101 – were the PAR scores and the four early warning sign scores shared with the intensivists who made the decisions about ICU admission on those occasions when the nurse first raised the alarm?

Response) We understand your concern. All of the scores mentioned in our study were not known to the intensivists who made the decisions about ICU admission. It was mentioned in the methods section as following:

“… The RRT nurses and physicians were not aware of these scores at the time of their visits and assessments, and the scores did not influence clinical decisions ….”

5. Line 124: please give more detail on the logistic regression methods i.e. how variables were selected, entered, and removed from, the final models.

Response) VitalPAC early warning score was selected to be combined with patient acuity rating, because it showed the largest area under the receiver operating characteristic curve compared to other warning scores (modified early warning score, standardised early warning score, and cardiac arrest risk triage). We agree that the previous description may mislead the readers, therefore we have revised the sentences as following:

“… Because PAR involves subjective assessment of patient status, the AUROCs of PAR for predicting the composite outcome according to each participating nurse were analyzed. A combined model of PAR with ViEWS, which showed the largest AUROC among the calculated scores, was built with logistic regression analysis, and the AUROC of this model was calculated as well. …”

6. Line 169: which statistical test was used to generate this probability level? None of the statistical tests described in the Methods seem appropriate. ANOVA or Kruskal-Wallis would be the tests usually employed for multiple group comparisons such as this.

Response) The areas under the receiver operating characteristic curve were compared using the DeLong method, with Bonferroni-adjustment. The information was added to the methods section as following:

“… The AUROCs for PAR and other scoring systems were compared using the DeLong method with Bonferroni-adjustment. …”

7. Adding some discussion of the similarities and differences between the authors’ PAR results and those of related studies (e.g. Edelson et al (2011) [ref 10] and O’Donnell et al (2016) [ref 11]) is important in order to place the findings in the context of the available literature - similar to the way the authors discuss their early warning score findings commencing at line 268.

Response) We appreciate your great idea. The study by Edelson et al (2011) was mentioned, which revealed the area under the receiver operating characteristic curve of patient acuity rating of 0.82, which was scored by physicians. 

“… Our results highlight that non-physician healthcare professionals are helpful in the RRT setting. A previous study has demonstrated the AUROC of PAR assessed by physicians to be 0.82 (0.69 for residents and 0.85 for attendings) for predicting short-term patient deterioration [11], which is comparable to the AUROC in this study (0.87 [95% CI 0.84–0.89]). …”

8. The authors should acknowledge the limitations of their data when discussing the importance of nurse work experience and the accuracy of PAR scores. Specifically, they identified two nurses with less RRT experience whose PAR score accuracy was compared with three more experienced RRT nurses. Such a small sample size (of nurses) cannot provide an adequate test of the impact of experience on PAR accuracy so greater caution in generalizing this finding is called for. The least experienced nurses might have had other characteristics which were of equal or greater relevance to the question of their PAR accuracy.

Response) We deeply agree with you on the issue. The small sample size of nurses (5 nurses) makes statistical analysis not viable. Although our study findings cannot be generalized to other nurses with diverse backgrounds, it may suggest that sufficient working experience as a RRT nurse is an important component of a nurse’s ability to correctly predict patient prognosis. A sentence in the discussion section was amended as following:

“… One of the major drawbacks of PAR is its subjective nature. As shown in our study, the reliability of PAR was questionable: PAR of nurses with longer experience in the RRT revealed an AUROC of over 0.9 in predicting mortality and/or ICU admission within the next day, but those with less experience (<6 months to 1 year) showed an AUROC of under 0.8. Although the small number of nurses makes statistical comparison impossible, it seems that sufficient work experience as an RRT nurse may be an important component of a nurse’s ability to correctly predict patient prognosis. …”

9. Lines 281-283: the authors rightly identify a crucial limitation of the design i.e. that the nurses involved were not independent assessors of patient acuity but involved in the triage process itself, which would likely inflate the AUROCs of the PAR scores in relation to the admission to ICU outcome. It would be helpful if the authors could reflect at this juncture on improved aspects of design which would provide a more robust test of nurse-scored PAR accuracy in this clinical context.

Response) Thank you for your careful reading. Future studies with blinded researchers may further assess the accuracy of PAR. A sentence was added to the discussion section mentioning the limitations as following:

“… Second, although final decision of patient management was made by the notified physician, the RRT nurse’s subjective opinion could have influenced the physician’s decisions, which might have led to more frequent ICU admissions for patients with high PAR. To exclude such potential influences, PAR needs to be assessed by nurses outside the RRT pathway in future studies. …”

MINOR COMMENTS

10. Lines 52-59: A slightly more comprehensive summary of the early warning score methods available would strengthen the Introduction – especially so if addressing the pros and cons of all four of the scores used in the study.

Response) We agree with your comment. Mentioning of other early warning scores was added to the introduction as following:

“To facilitate early detection of deteriorating patient status, various early warning scores have been developed. For example, the modified early warning Score (MEWS), one of the most well-known early warning scores, is used in several clinics to facilitate decision-making [2, 4, 5]. However, MEWS provides variable accuracy, and it remains unclear whether the score can be used alone to predict unexpected critical events [3, 6]. Other scoring systems, including VitalPAC early warning score (ViEWS) [7], standardised early warning score (SEWS) [8], and the cardiac arrest risk triage (CART), have also been introduced; however, they were revealed to have performance similar to that of MEWS [3]. These factors have led to the development of several machine learning methods and new risk stratification tools, although the complexity of these techniques can make it difficult to implement these in a pragmatic manner [9, 10].”

11: The authors chose to compare the AUROC of the PAR, a subjective measure, with four objective measures derived from clinical signs. Why did they choose these particular objective measures (see Comment 10)? Did they explore the possibility of making comparisons between PAR and other subjective measures of patient acuity? Explanation of the rationale in both cases would be informative in assessing the validity of the design.

Response) Thank you for your careful reading. The main aim of this study was to assess the performance of subjective assessment of the patients by RRT nurses. In order to do so, we decided to compare patient acuity rating against other well-known early warning scores, including modified early warning score, VitalPAC early warning score, standardised early warning score, and cardiac arrest risk triage. Comparing patient acuity rating against other subjective measures was not appropriate, because the main aim of our study was to assess the performance of the subjective measure, which has not been studied before.

“During the study period, the RRT nurses evaluated the probability of patient deterioration using the PAR, from a scale of 1 to 7 [11]. PAR of 1 corresponds to the lowest probability of patient deterioration and 7 corresponds to the highest. A diagram for better understanding of PAR is available from the original study [11]. Other well-known scores for predicting short-term patient deterioration, including MEWS, ViEWS, SEWS, and the CART score, were calculated afterwards.”

12. Line 131: “Patterns of distributions were assessed by visual inspection” – what was the rationale for not testing the normality and heterogeneity of variance of the distributions using commonly used tests?

Response) We appreciate your comment. After Shapiro-Wilk test, “Age” and “Body mass index” revealed not to follow the normal distribution. Therefore, the descriptions were changed into median and interquartile ranges throughout the manuscript.

13. Line 139: “composite outcome” – please clarify the meaning of this phrase at this point.

Response) Thank you for your concern. The sentence was amended as following:

“During the study period, 1,441 patients triggered the RRT. Nine patients with sudden cardiac arrest and six patients whom PAR was not assessed were excluded. Therefore, 1,426 patients were included in the final analysis. Among them, 258 patients (18.1%) experienced death and/or ICU admission within 1 day, defined as the “composite outcome”.”

14. Line 146: “when the RRT was contacted directly” – please explain what this means – what were the other means by which the RRTs were engaged?

Response) We apologize for the ambiguous description. It meant that they were contacted directly by phone because of concerns other than specified abnormalities of vital signs and laboratory findings. The sentence was amended as following:

“Composite outcome was expected when patients had abnormalities in serum pH (3.9% vs. 0.6%, P<0.001) or when the RRT was contacted directly by phone due to other concerns (18.2% vs. 4.5%, P<0.001).”

15. Line 169: Please provide the probabilities associated with the analyses of differences in AUROC’s between the five measures for the ICU admission and mortality outcomes.

Response) As recommended, we added the probabilities associated with the analyses into the main manuscript. 

“PAR was superior in anticipating ICU admission (AUROC, 0.87 [95% CI, 0.84–0.89]) than MEWS (AUROC, 0.65 [95% CI, 0.61–0.69]), ViEWS (AUROC, 0.68 [95% CI 0.64–0.72]), SEWS (AUROC, 0.66 [95% CI 0.62–0.70]), and CART (AUROC, 0.63 [95% CI 0.59–0.67]) but was not superior in predicting mortality (AUROC, 0.79 [95% CI 0.70–0.87]) than MEWS (AUROC, 0.69 [95% CI 0.61–0.76]), ViEWS (AUROC, 0.70 [95% CI 0.61–0.78]), SEWS (AUROC, 0.69 [95% CI 0.62–0.77]), and CART (AUROC, 0.58 [95% CI 0.49–0.67]) (Table 2).”

16. Lines 193-196 and Lines 200-202: Please quote the probabilities associated with these three analyses.

Response) As recommended, we added the probabilities associated with the analyses into the main manuscript. 

“The combined model of PAR and ViEWS in the overall population showed a significantly improved AUROC (0.875 [95% CI, 0.849–0.900]) compared to that of PAR alone (0.868 [95% CI 0.843–0.894]). This change was mainly due to the improvement in AUROC for nurse 4; the combined model showed a significantly improved AUROC (0.81 [95% CI 0.75–0.86]) for predicting patient outcome compared to that of PAR alone (0.78 [95% CI 0.72–0.84]).”

17. The analyses by individual nurses, taking into consideration their levels of experience, and the logistic regression modelling of the predictive value of PAR and ViEWS in combination bring added value to the design and are of practical significance.

Response) We appreciate your comment. 

18. Are the sentences at lines 215 and 223 not contradictory?

Response) Sentence at line 215 tried to emphasize that this was the first study to be performed in the “RRT setting.” Sentence at line 223 tried to mention other studies performed in other clinical settings. To reduce the possibility of misleading the readers, we have amended the sentences as following:

“This is the first study to demonstrate the superiority of PAR over other early warning scores in predicting short-term patient prognosis in the RRT setting, highlighting the importance of bedside patient evaluation implemented in previous studies with different clinical settings [16, 17].”

 

Reviewer #3: 

Thank you for allowing me to review this study by Kim et al. Rapid response teams are a major development in inpatient care and understanding better ways to objectively triage and assess patients is an important issue. I also think the more we can understand how all providers of care (nurses, physical therapists, physicians) interact with patients is extremely useful. Thus, I read this study with great interest.

The authors used a retrospective review of a prospective observational cohort of patients at a single health center and how patients were assessed by rapid response team nurses. They reported finding that a subjective, 7-point Likert scale, the PAR performed better overall than other, more objective scores. The authors also spent a lot of time dissecting the experience of one less-experienced team member versus the other four.

I think that there is some interest in the overall finding - PAR performed better than other scores. I am very guarded in my review of the study however because the authors used almost half of the manuscript to discuss essentially a descriptive difference between 2 nurses newer to the team to 3 others who had been on the team for a longer period of time. This study was not designed to assess the experience of the nurses on a rapid response team (statistics cannot be performed this low of a sample size). I thus think this study presentation needs to be rebuilt around the actual findings of the study.

Major issues:

1. There is no power to assess for the experience of nurses and the authors report experience of nurses as a major conclusion of the entire study

There are 3 nurses with >4 years’ experience as an RRT nurse. There are 2 with 1 year or less experience. Only 1 of 2 nurses with less experience (Nurse 4) had a significantly lower AUROC than the 3 more experienced nurses (Nurses 1-3). There is no way to make even a subjective conclusion off a single individual underperforming (Nurse 4) and the other individual performing the same (Nurse 5). I don’t even think it belongs in the manuscript beyond perhaps a description of the 5 nurses in the RRT team. I would draw the conclusion the nurses performed the same except for one. They also had less training, but another nurse with even less training (Nurse 5) performed the same.

Because this is reported as a major conclusion of the study throughout, including the abstract, this entire manuscript needs to be refocused on the actual objective study findings before it can be fully vetted.

Response) We deeply understand your concern. Although the experience of the assessing RRT nurse may be an import factor for higher performance of PAR, we agree that small number of nurses in this study makes it difficult to include the findings as a major conclusion. After careful discussion between the authors, we have decided to delete the sentences about experience of nurses as a major conclusion from both the abstract and the main text. The text was amended as following:

“Conclusions: PAR assessed by RRT nurses can be a useful tool for assessing short-term patient prognosis in the RRT setting. 

…

Conclusions

In conclusion, subjective assessment of the patient by the RRT nurse, represented as PAR reveals good performance in predicting patient prognosis. Although early warning scores may be useful for identifying at-risk patients, direct examinations by healthcare professionals should be emphasized when RRT is activated.”

2. I am not sure the comparison scores are the best available

The authors admit that MEWS was not developed for general ward patients (it was developed for acute medical admissions). The scores also looked at different outcomes. Why did the authors choose these scores? Is it fair to say the PAR score outperforms scores that are not clearly validated in the actual patient population? Is this important? ViEWS and CART seem fair, but SEWS, and most certainly MEWS, seem less applicable. Are there better scores to compare? The authors calculate the Charlton Comorbidity Index as important but did not compare the PAR with this score. I think it would be helpful.

Response) We appreciate your comment. Although the scores were created from diverse clinical settings, the scores used in our study (MEWS, ViEWS, SEWS, and CART) were meant to anticipate short-term patient prognosis. Furthermore, the area under the receiver operating characteristic curve (AUROC) of patient acuity rating (PAR) was calculated to be over 0.8 in our study, which refers to good power of prediction by itself.

Charlson Comorbidity Index (CCI), on the other hand, is meant to predict 10-year mortality. Therefore, comparing CCI against PAR does not seem to be appropriate. The area under the receiver operating characteristic curve of CCI is calculated to be 0.485 in our population, to predict mortality and/or ICU admission within the next day. 

Sentences were added to the manuscript as following in the discussion section: 

“Our study identified smaller AUROC values for other early waring scores compared to previous studies [3, 4, 7, 14, 15]. This is likely related to the studies being performed in different medical settings. MEWS was validated in the acute medical admissions setting, and nurses collected the scoring variables during their routine duties [4]. ViEWS and SEWS were studied in the emergency treatment setting [7, 8], and CART was validated in the general ward [15]. Moreover, these tools targeted different clinical outcomes in each studies: admission to a higher dependency unit, attendance of the cardiac arrest team, death and survival at 60 days (MEWS) [4]; in-hospital mortality within 24 hours (ViEWS) [7]; in-hospital mortality and length of stay (SEWS) [8]; and in-hospital cardiac arrest and ICU admission (CART) [15]. These differences could create a noticeable difference in the AUROC values produced, as our study calculated AUROC with selected patients who activated the RRT, in which worse prognosis is anticipated compared to the normal patient population. However, this does not devaluate the power of PAR in our study; the AUROC of PAR was 0.87 (95% CI 0.84–0.89), which refers to good power for predicting short-term patient deterioration.”

3. What was the best cut-off to predict ICU admission for the PAR in this study?

I was a bit confused that the PAR score is deemed more reliable a determination for predicting bad outcomes in patients who have an RRT consultation, however, it was not expressly clear to me what cut-off was used to predict bad outcome most accurately. Certainly, if one is to apply the use of PAR for clinical practice, there should be some cutoff point for providers to follow as to should the patient be admitted to the ICU. The authors need to discuss this as it is critical for actual applying a diagnostic test.

Response) We appreciate your careful reading. A cut-off value of 4 may be a reasonable point with good sensitivity (84.9%) and fair specificity (73.2%) for predicting short-term patient deterioration. It was added to the discussion section as following:

“Our results highlight that non-physician healthcare professionals are helpful in the RRT setting. A previous study has demonstrated the AUROC of PAR assessed by physicians to be 0.82 (0.69 for residents and 0.85 for attendings) for predicting short-term patient deterioration [11], which is comparable to the AUROC in this study (0.87 [95% CI 0.84–0.89]). Recent reductions in resident duty hours (80-hour maximum weekly limit) by the Accreditation Council for Graduate Medical Education have created longer handoff periods between inpatient physicians [20], which leads to an increased need for non-physician healthcare professionals. In combination with early warning scores, the use of subjective patient assessments by ward nurses can provide a better ability to predict patient deterioration [21]. Considering the sensitivity and specificity of PAR (Table 3), a cut-off value of ≥4 can be reasonable, with good sensitivity and fair specificity for predicting short-term patient deterioration.”

4. One combination assessment was tested – PAR + ViEWS – but not others

It was reported in Table 4 about the individual nurse subjects’ performance on PAR, ViEWS and PAR+ViEWS. As stated above, the comparison between nurses for experience is a minor point. It would be much more useful to assess if combining the PAR with the ViEWS in the overall cohort was superior to either alone.

Response) We appreciate your comment. We compared PAR versus PAR+ViEWS in the overall population. PAR+ViEWS reveal larger area under the receiver operating characteristic curve compared to PAR alone, but this is mainly due to “nurse 4.” The difference was only significant in patients assessed by “nurse 4,” but not in patients assessed by other nurses. Therefore, the superiority of PAR+ViEWS was due to “nurse 4,” rather than other nurses. The findings were added to the results section as following:

“To compensate for the smaller AUROC of less-experienced nurses’ PAR, we utilized a logistic regression model of PAR with the warning score of the largest AUROC in our study: ViEWS. The combined model of PAR and ViEWS in the overall population showed a significantly improved AUROC (0.875 [95% CI, 0.849–0.900]) compared to that of PAR alone (0.868 [95% CI 0.843–0.894]). This change was mainly due to the improvement in AUROC for nurse 4; the combined model showed a significantly improved AUROC (0.81 [95% CI 0.75–0.86]) for predicting patient outcome compared to that of PAR alone (0.78 [95% CI 0.72–0.84]). Meanwhile, significant improvement of AUROC in this model was not observed for the other four nurses (Table 4).”

5. Authors draw conclusions about nurses in general, but only tested five RRT nurses and did not compare performance to other providers in the same cohort of patients

The authors draw many conclusions on nurses’ overall assessments versus the performance of other providers (physicians, for example), however this is a highly trained and experienced subset of nurses. There was also no comparison between different types of providers (advanced practice providers, interns, residents, attending physicians) in this particular cohort of patients. To truly make a comparison of these types of providers ability to assess the PAR accurately, their performance on the PAR would need to examined in the same cohort of patients. I think minimal conclusions in direct comparison can be drawn from this study. I also think generalizing the performance of RRT nurses to all nurses throughout the hospital, with a wide subset of skills and specialties, is impossible.

Response) We admit the limitation of our study, and agree that our findings may not be generalized to other less-experienced nurses. With help of a qualified statistician, the reliability of PAR in predicting the composite outcome was assessed with calibration plot and Hosmer-Lemeshow test, which revealed poor reliability. Patient acuity rating was assessed only by RRT nurses in our study, therefore, comparison between other providers was not possible. We have added several sentences to the limitation, and have made careful modifications to the conclusion as following:

“This study has several limitations. First, this study was a single-center design which includes only 5 RRT nurses with long working experience, and our results may not be generalized to other less experienced nurses. Inter-observer variability should be considered, and further studies on healthcare professionals with various backgrounds can enforce the strength of PAR. Second, although final decision of patient management was made by the notified physician, the RRT nurse’s subjective opinion could have influenced the physician’s decisions, which might have led to more frequent ICU admissions for patients with high PAR. To exclude such potential influences, PAR needs to be assessed by nurses outside the RRT pathway in future studies. Third, this study included patients who have certain extent of systemic illness. This may influence the AUROC of PAR and early warning scores. 

…

In conclusion, subjective assessment of the patient by the RRT nurse, represented as PAR reveals good performance in predicting patient prognosis. Although early warning scores may be useful for identifying at-risk patients, direct examinations by healthcare professionals should be emphasized when RRT is activated.”

6. Were any further outcomes assessed? If ICU admission at a later time is an outcome, what was the overall outcome of the patients?

Not all ICU admissions are appropriate and more evidence is coming out that perhaps needless ICU admissions are without harm. I also am interested that mortality was actually not significantly predicted by the PAR, which might suggest the ICU admissions did not correlate with mortality as much. Was there any more data to be gained? I would be very interested in overall in-patient mortality. Additionally, if available, 6 month and 1-year mortality would be very useful, although I realize this may be difficult to obtain.

Response) We appreciate your idea. However, the data of 6-month and 1-year mortality could not be obtained. The main aim of our study was to anticipate short-term patient prognosis.

Minor issues:

1. In table 1, need to put sex, female statistics (how many total, death/ICU admission, alive without ICU admission within 1 day)

Response) We have added female statistics into Table 1. 

2. What was the p value comparing in sex, age, BMI Charlson comorbidity index in Table 1? Was it if that variable predicted outcome on logistic regression? State this in the table details.

Response) The p-values were added to the manuscript. After logistic regression analysis, BMI was associated with the composite outcome (odds ratio 0.97, 95% confidence interval 0.94–1.00, per kg/m2 of BMI, P-value = 0.044). However, BMI was not known to the RRT nurse at the time of PAR assessment, therefore did not influence PAR.

“The patients exhibited male predominance (60.6%), a median age of 72 (IQR, 61–79) years, a mean body mass index of 21.4 (IQR, 19.4–24.7) kg/m2, and a median Charlson comorbidity index of 2 (IQR, 1–4). Patients who died and/or admitted to the ICU seemed to have lower body mass index than those who did not (median 21.2 vs. 22.0, P-value=0.020). Sex, age, and Charlson comorbidity index did not differ significantly between the two groups (P-values 0.173, 0.310, and 0.427, respectively).”

3. Table 1, don’t need to state detailed criteria for rapid response team in the caption

Response) We agree with your comment. The caption was deleted.

4. The details of how the RRT work at the hospital is long and extremely detailed. I’m not sure we need all the shift times.

Response) Thank you for your comment. We have deleted the redundant sentences, and have amended the sentences in the methods section as following:

“This study included adult patients who required RRT support during their admission between November 1, 2016, and December 12, 2017. ...

Sample size was calculated by estimating the area under the receiver operating characteristic curve (AUROC) of PAR as 0.78, according to a pilot analysis of patients from January 2015 to March 2016 in our center, and the AUROC of MEWS as 0.72, according to a previous report [4]. The RRT was activated by the following criteria: systolic blood pressure <90 mmHg; heart rate <50/min or >140/min; respiratory rate <10/min or >30/min; body temperature >39 °C or <36 °C; peripheral oxygen saturation <90% with room air and/or facial mask with oxygen flow >8 L/min; serum pH <7.3; serum partial pressure of carbon dioxide >50 mmHg; serum partial pressure of oxygen <60 mmHg; serum lactic acid >2.5 mmol/L; serum total carbon dioxide <15 mmol/L; or direct concerns from ward nurses. Patients with sudden cardiac arrest and those in whom PAR was not assessed were excluded from the study. In patients with multiple instances of RRT support, only the first instance was included in the analysis.

Data of baseline demographics, Charlson comorbidity index, department of admission, causes of RRT notification, characteristics of RRT triggering, result of the RRT intervention, and RRT nurse-assessed PAR for the probability of ICU admission and/or mortality within the next day were collected prospectively. 

…

Patient assessment and score calculation

During the study period, the RRT nurses evaluated the probability of patient deterioration using the PAR, from a scale of 1 to 7 [11]. PAR of 1 corresponds to the lowest probability of patient deterioration and 7 corresponds to the highest. A diagram for better understanding of PAR is available from the original study [11]. Other well-known scores for predicting short-term patient deterioration, including MEWS, ViEWS, SEWS, and the CART score, were calculated afterwards. 

…

Nurses in the RRT

Five nurses, who were co-authors of our study, were included in the RRT. All nurses had >9 years of experience of working as a nurse and at least 5 years of experience in the ICU. Nurses 1, 2, and 3 had a >4-year experience of working in the RRT whereas nurses 4 and 5 had a <2-year experience of working in the RRT. Nurses 1, 2, 3, and 4 had completed an ICU nursing program, a nurse preceptor program, and a basic life support provider program. Nurses 1 and 2 had completed an advanced cardiac life support provider program, and nurse 2 was working as an instructor for a basic life support provider program. The RRT nurses did not have direct authority on the final medical decision.”

5. What is a part-time intensivist? Is this a critical care attending physician? Is it a moon-lighter in another specialty? The term is imprecise and I do not know what it means. Also, there was mention of ICU fellows, but in what subspecialty? Critical care medicine?

Response) Part-time intensivist refers to physicians who participate in the RRT only part-time. They attend to other clinical work such as outpatient clinic when they are off-shift. ICU fellows refer to clinical fellows participating in the management of ICU patients, and they specialize in pulmonology, nephrology, and emergency medicine. Such descriptions seemed redundant, therefore were deleted from the manuscript.

6. When were clinical variables for the various scores obtained? At bedside or last vital sign check? Same day labs or admission labs?

Response) The clinical variables for the scores were obtained from the latest alerted vital signs and/or lab findings upon RRT activation, which was on the same day. 

“Details of each score calculation are available in the Tables S1–S4 of the supporting information. Patients’ vital signs used in score calculation were recorded by ward nurses; these data were immediately sent to the RRT. The RRT nurses and physicians were not aware of these scores at the time of their visits and assessments, and the scores did not influence clinical decisions.”

7. Imprecise language in discussion paragraph 3 (page 12). “the possibility that non-physician healthcare professionals can be helpful in the RRT setting”. Certainly no one would argue that a non-physician healthcare professional is not helpful in an RRT. I don’t understand what the point of the statement is but it could be construed as hurtful to imply the care team is not helpful.

Response) We agree about the imprecise description. The sentence was amended as following:

“Our results highlight that non-physician healthcare professionals are helpful in the RRT setting.”

8. Discussion paragraph 3, page 12: ACGME duty hours is not referenced. I thought that the ACGME went back to allowing 28-hour shifts for interns. There needs to be an accurate depiction of what the policy is with up to date citations.

Response) According to the 2017 ACGME common program requirements, a maximum working limit of 80 hours (exceptions for up to 88 hours if educational work was included) is mandatory. Sentences were amended, and appropriate up to date citation was added.

“Recent reductions in resident duty hours (80-hour maximum weekly limit) by the Accreditation Council for Graduate Medical Education have created longer handoff periods between inpatient physicians [20], which leads to an increased need for non-physician healthcare professionals.”

---

## [Decision Letter · Decision Letter 1]

13 Sep 2019

PONE-D-19-17205R1

Performance of Patient Acuity Rating by Rapid Response Team Nurses for Predicting Short-Term Prognosis

PLOS ONE

Dear Yeon Joo Lee,

Thank you for submitting your manuscript to PLOS ONE. After careful consideration, we feel that it has merit but does not fully meet PLOS ONE’s publication criteria as it currently stands. Therefore, we invite you to submit a revised version of the manuscript that addresses the points raised during the review process.

ACADEMIC EDITOR: Our expert reviewers still have raised a number of points which we believe major modifications are necessary to improve the manuscript, taking into account the reviewers' remarks below.

We would appreciate receiving your revised manuscript by Oct 28 2019 11:59PM. To enhance the reproducibility of your results, we recommend that if applicable you deposit your laboratory protocols in protocols.io, where a protocol can be assigned its own identifier (DOI) such that it can be cited independently in the future. For instructions see: http://journals.plos.org/plosone/s/submission-guidelines#loc-laboratory-protocols

We look forward to receiving your revised manuscript.

Kind regards,

Wisit Cheungpasitporn, MD, FACP

University of Mississippi Medical Center

Twitter: @wisit661 Email: wcheungpasitporn@gmail.com 

Academic Editor

PLOS ONE

Reviewers' comments:

Reviewer's Responses to Questions

**Comments to the Author**

1. If the authors have adequately addressed your comments raised in a previous round of review and you feel that this manuscript is now acceptable for publication, you may indicate that here to bypass the “Comments to the Author” section, enter your conflict of interest statement in the “Confidential to Editor” section, and submit your "Accept" recommendation.

Reviewer #1: All comments have been addressed

Reviewer #2: (No Response)

Reviewer #3: (No Response)

2. Is the manuscript technically sound, and do the data support the conclusions?

Reviewer #1: Yes

Reviewer #2: Partly

Reviewer #3: Partly

3. Has the statistical analysis been performed appropriately and rigorously? 

Reviewer #1: Yes

Reviewer #2: No

Reviewer #3: Yes

4. Have the authors made all data underlying the findings in their manuscript fully available?

Reviewer #1: Yes

Reviewer #2: Yes

Reviewer #3: Yes

5. Is the manuscript presented in an intelligible fashion and written in standard English?

Reviewer #1: Yes

Reviewer #2: Yes

Reviewer #3: Yes

6. Review Comments to the Author

Reviewer #1: Thanks for putting the needed effort to enhance your manuscript and make it sounding with a good message to the readers

Reviewer #2: OVERALL IMPRESSION

The authors have addressed the majority of my comments relating to the previous submission in appropriate ways and this has significantly improved the manuscript, especially in terms of clarity regarding the methods and greater caution in their discussion. However, they have not adequately addressed four of my previous comments, one of which I regard as a major issue

MAJOR COMMENTS

Materials and Methods

Comment 1: Lines 125-132: The response to my original comment “Please give more detail on the logistic regression methods i.e. how variables were selected, entered, and removed from, the final models” has not been adequately addressed. It remains unclear which variables were considered for inclusion in the logistic regression modelling of the composite outcome, and the rationale behind these considerations.

MINOR COMMENTS

Results

Comment 2: Lines 174-179: Regarding the response to my original comment to “Please provide the probabilities associated with the analyses of differences in AUROC’s between the five measures for the ICU admission and mortality outcomes” the authors have added the estimates and CI’s of the individual early warning sign methods which is to be welcomed – but they have not added the probability levels for either of these tests of group differences in AUROC as they have done for the composite outcome dependent variable.

Comment 3: Lines 208-220: Similarly, regarding the response to my original comment “Please quote the probabilities associated with these three analyses” the authors have provided the estimates and CI’s but not the p values.

Discussion

Comment 4: Sentence commencing at Line 272: Regarding the response to my original comment “The authors should acknowledge the limitations of their data when discussing the importance of nurse work experience and the accuracy of PAR scores. Specifically, they identified two nurses with less RRT experience whose PAR score accuracy was compared with three more experienced RRT nurses. Such a small sample size (of nurses) cannot provide an adequate test of the impact of experience on PAR accuracy so greater caution in generalising this finding is called for. The least experienced nurses might have had other characteristics which were of equal or greater relevance to the question of their PAR accuracy” the authors have made improvements to the text to qualify this area of their analysis. They should go further by adding at the end of the sentence commencing at Line 272 a further clause such as “...although this would need to be tested in appropriately designed future studies”

Reviewer #3: Review for “Patient Acuity Rating by Nurses in Rapid Response Team”

Thank you for allowing me to re-review this study by Kim et al. Rapid response teams are a major development in inpatient care and understanding better ways to objectively triage and assess patients is an important issue. I also think the more we can understand how all providers of care (nurses, physical therapists, physicians) interact with patients is extremely useful.

The authors took a lot of effort to respond to the comments of reviewers. That said, there are still a lot of limitations to the scope and design of the study.

I think my biggest issue still is PAR versus PAR+ViEWS. The authors have found that there is poor reliability in the PAR score. The sample size of nurses is extremely small. Eliminating the single nurse with less experience is not possible in this analysis. If experience is to be tested, you need to expand the study with more nurses and report the results afterwards. Otherwise, it is a subjective study point. Your overall statistical conclusion is the PAR+ViEWS is superior to the PAR. I address the author responses to my comments, however, addressing major issue #4 requires a major revision on conclusions and the overall manuscript. The fact that PAR+ViEWS is superior to PAR needs to be emphasized and discussed. It is not even mentioned in the abstract.

Major issues:

1) ORIGINAL COMMENT: There is no power to assess for the experience of nurses and the authors report experience of nurses as a major conclusion of the entire study

There are 3 nurses with >4 years experience as an RRT nurse. There are 2 with 1 year or less experience. Only 1 of 2 nurses with less experience (Nurse 4) had a significantly lower AUROC than the 3 more experienced nurses (Nurses 1-3). There is no way to make even a subjective conclusion off a single individual underperforming (Nurse 4) and the other individual performing the same (Nurse 5). I don’t even think it belongs in the manuscript beyond perhaps a description of the 5 nurses in the RRT team. I would draw the conclusion the nurses performed the same except for one. They also had less training, but another nurse with even less training (Nurse 5) performed the same.

Because this is reported as a major conclusion of the study throughout, including the abstract, this entire manuscript needs to be refocused on the actual objective study findings before it can be fully vetted.

Response) We deeply understand your concern. Although the experience of the

assessing RRT nurse may be an import factor for higher performance of PAR, we

agree that small number of nurses in this study makes it difficult to include the findings

as a major conclusion. After careful discussion between the authors, we have decided

to delete the sentences about experience of nurses as a major conclusion from both

the abstract and the main text. The text was amended as following:

“Conclusions: PAR assessed by RRT nurses can be a useful tool for assessing shortterm

patient prognosis in the RRT setting.

…

Conclusions

In conclusion, subjective assessment of the patient by the RRT nurse, represented as

PAR reveals good performance in predicting patient prognosis. Although early warning

scores may be useful for identifying at-risk patients, direct examinations by healthcare

professionals should be emphasized when RRT is activated.”

NEW COMMENT: I think that the authors have de-emphasized the experience factor, however, see comments above. If you truly de-emphasize the splitting of one nurse, your study conclusion changes = PAR+ViEWS is superior to PAR.

2) ORIGINAL COMMENT: I am not sure the comparison scores are the best available

The authors admit that MEWS was not developed for general ward patients (it was developed for acute medical admissions). The scores also looked at different outcomes. Why did the authors choose these scores? Is it fair to say the PAR score outperforms scores that are not clearly validated in the actual patient population? Is this important? ViEWS and CART seem fair, but SEWS, and most certainly MEWS, seem less applicable. Are there better scores to compare? The authors calculate the Charlton Comorbidity Index as important but did not compare the PAR with this score. I think it would be helpful.

Response) We appreciate your comment. Although the scores were created from

diverse clinical settings, the scores used in our study (MEWS, ViEWS, SEWS, and

CART) were meant to anticipate short-term patient prognosis. Furthermore, the area under the receiver operating characteristic curve (AUROC) of patient acuity rating (PAR) was calculated to be over 0.8 in our study, which refers to good power of prediction by itself.

Charlson Comorbidity Index (CCI), on the other hand, is meant to predict 10-year

mortality. Therefore, comparing CCI against PAR does not seem to be appropriate.

The area under the receiver operating characteristic curve of CCI is calculated to be

0.485 in our population, to predict mortality and/or ICU admission within the next day.

Sentences were added to the manuscript as following in the discussion section:

“Our study identified smaller AUROC values for other early waring scores compared to

previous studies [3, 4, 7, 14, 15]. This is likely related to the studies being performed in

different medical settings. MEWS was validated in the acute medical admissions

setting, and nurses collected the scoring variables during their routine duties [4].

ViEWS and SEWS were studied in the emergency treatment setting [7, 8], and CART

was validated in the general ward [15]. Moreover, these tools targeted different clinical

outcomes in each studies: admission to a higher dependency unit, attendance of the

cardiac arrest team, death and survival at 60 days (MEWS) [4]; in-hospital mortality

within 24 hours (ViEWS) [7]; in-hospital mortality and length of stay (SEWS) [8]; and inhospital

cardiac arrest and ICU admission (CART) [15]. These differences could create

a noticeable difference in the AUROC values produced, as our study calculated

AUROC with selected patients who activated the RRT, in which worse prognosis is

anticipated compared to the normal patient population. However, this does not

devaluate the power of PAR in our study; the AUROC of PAR was 0.87 (95% CI

0.84–0.89), which refers to good power for predicting short-term patient deterioration.”

NEW COMMENT: I think that mentioning the limitations of the other methods, and how they are not validated in this population is enough at this point.

I think the authors missed my point on the Charlson Comorbidity Index. If the authors are calculating the CCI as an important feature, it was originally designed as a diagnostic algorithm to predict outcome. However, as they do not have outcome data available as stated, then I agree there is no point in adding this as a comparison.

The PAR+ViEWS is superior to all of these scores?

3) ORIGINAL COMMENT: What was the best cut-off to predict ICU admission for the PAR in this study?

I was a bit confused that the PAR score is deemed more reliable a determination for predicting bad outcomes in patients who have an RRT consultation, however, it was not expressly clear to me what cut-off was used to predict bad outcome most accurately. Certainly, if one is to apply the use of PAR for clinical practice, there should be some cutoff point for providers to follow as to should the patient be admitted to the ICU. The authors need to discuss this as it is critical for actual applying a diagnostic test.

Response) We appreciate your careful reading. A cut-off value of 4 may be a

reasonable point with good sensitivity (84.9%) and fair specificity (73.2%) for predicting

short-term patient deterioration. It was added to the discussion section as following:

“Our results highlight that non-physician healthcare professionals are helpful in the

RRT setting. A previous study has demonstrated the AUROC of PAR assessed by

physicians to be 0.82 (0.69 for residents and 0.85 for attendings) for predicting shortterm

patient deterioration [11], which is comparable to the AUROC in this study (0.87

[95% CI 0.84–0.89]). Recent reductions in resident duty hours (80-hour maximum

weekly limit) by the Accreditation Council for Graduate Medical Education have created

longer handoff periods between inpatient physicians [20], which leads to an increased

need for non-physician healthcare professionals. In combination with early warning

scores, the use of subjective patient assessments by ward nurses can provide a better

ability to predict patient deterioration [21]. Considering the sensitivity and specificity of

PAR (Table 3), a cut-off value of ≥4 can be reasonable, with good sensitivity and fair

specificity for predicting short-term patient deterioration.”

NEW COMMENT: This is clearer to me now

4) ORIGINAL COMMENT: One combination assessment was tested – PAR + ViEWS – but not others

It was reported in Table 4 about the individual nurse subjects’ performance on PAR, ViEWS and PAR+ViEWS. As stated above, the comparison between nurses for experience is a minor point. It would be much more useful to assess if combining the PAR with the ViEWS in the overall cohort was superior to either alone.

Response) We appreciate your comment. We compared PAR versus PAR+ViEWS in

the overall population. PAR+ViEWS reveal larger area under the receiver operating

characteristic curve compared to PAR alone, but this is mainly due to “nurse 4.” The

difference was only significant in patients assessed by “nurse 4,” but not in patients

assessed by other nurses. Therefore, the superiority of PAR+ViEWS was due to “nurse 4,” rather than other nurses. The findings were added to the results section as

following:

“To compensate for the smaller AUROC of less-experienced nurses’ PAR, we utilized a

logistic regression model of PAR with the warning score of the largest AUROC in our

study: ViEWS. The combined model of PAR and ViEWS in the overall population

showed a significantly improved AUROC (0.875 [95% CI, 0.849–0.900]) compared to

that of PAR alone (0.868 [95% CI 0.843–0.894]). This change was mainly due to the

improvement in AUROC for nurse 4; the combined model showed a significantly

improved AUROC (0.81 [95% CI 0.75–0.86]) for predicting patient outcome compared

to that of PAR alone (0.78 [95% CI 0.72–0.84]). Meanwhile, significant improvement of

AUROC in this model was not observed for the other four nurses (Table 4).”

NEW COMMENT: I disagree. Not every facility has nurses with a large amount of experience in an RRT team (in fact all of the nurses were quite experienced overall). There is such a small sample size that eliminating one of the nurses because they did not score similar to the others is not possible here. The PAR+ViEWS was superior to the PAR. The risk of n=5 is you have outliers; conversely, your group with a large amount of direct RRT-experienced nurses is probably less like RRT teams around the world – where there could be very inexperienced nurses. Again, this is the risk you run with your trial. The fact that if you take out the inexperienced nurse is a small discussion point, not a major conclusion.

5) ORIGINAL COMMENT: Authors draw conclusions about nurses in general, but only tested five RRT nurses and did not compare performance to other providers in the same cohort of patients

The authors draw many conclusions on nurses overall assessments versus the performance of other providers (physicians, for example), however this is a highly trained and experienced subset of nurses. There was also no comparison between different types of providers (advanced practice providers, interns, residents, attending physicians) in this particular cohort of patients. To truly make a comparison of these types of providers ability to assess the PAR accurately, their performance on the PAR would need to examined in the same cohort of patients. I think minimal conclusions in direct comparison can be drawn from this study. I also think generalizing the performance of RRT nurses to all nurses throughout the hospital, with a wide subset of skills and specialties, is impossible.

Response) We admit the limitation of our study, and agree that our findings may not be

generalized to other less-experienced nurses. With help of a qualified statistician, the

reliability of PAR in predicting the composite outcome was assessed with calibration

plot and Hosmer-Lemeshow test, which revealed poor reliability. Patient acuity rating

was assessed only by RRT nurses in our study, therefore, comparison between other

providers was not possible. We have added several sentences to the limitation, and

have made careful modifications to the conclusion as following:

“This study has several limitations. First, this study was a single-center design which

includes only 5 RRT nurses with long working experience, and our results may not be

generalized to other less experienced nurses. Inter-observer variability should be

considered, and further studies on healthcare professionals with various backgrounds

can enforce the strength of PAR. Second, although final decision of patient

management was made by the notified physician, the RRT nurse’s subjective opinion

could have influenced the physician’s decisions, which might have led to more frequent

ICU admissions for patients with high PAR. To exclude such potential influences, PAR

needs to be assessed by nurses outside the RRT pathway in future studies. Third, this

study included patients who have certain extent of systemic illness. This may influence

the AUROC of PAR and early warning scores.

…

In conclusion, subjective assessment of the patient by the RRT nurse, represented as

PAR reveals good performance in predicting patient prognosis. Although early warning

scores may be useful for identifying at-risk patients, direct examinations by healthcare

professionals should be emphasized when RRT is activated.”

NEW COMMENT: I would also state this in limitations – that you did not collect direct data on the PAR used for final triage by the physician/and or final decision maker.

6)ORIGINAL COMMENT: Were any further outcomes assessed? If ICU admission at a later time is an outcome, what was the overall outcome of the patients?

Not all ICU admissions are appropriate and more evidence is coming out that perhaps needless ICU admissions are without harm. I also am interested that mortality was actually not significantly predicted by the PAR, which might suggest the ICU admissions did not correlate with mortality as much. Was there any more data to be gained? I would be very interested in overall in-patient mortality. Additionally, if available, 6 month and 1 year mortality would be very useful, although I realize this may be difficult to obtain.

Response) We appreciate your idea. However, the data of 6-month and 1-year mortality could not be obtained. The main aim of our study was to anticipate short-term

patient prognosis.

NEW COMMENT: This should be mentioned in the limitation section, that we do not have this data and ICU admission as an outcome does not account for inappropriate/unneeded ICU admissions. Really, the outcome that is more important is mortality and functional status.

Minor issues:

1) In table 1, need to put sex, female statistics (how many total, death/ICU admission, alive without ICU admission within 1 day)

Response) We have added female statistics into Table 1.

CORRECTED

2) What was the p value comparing in sex, age, BMI Charlson comorbidity index in Table 1? Was it if that variable predicted outcome on logistic regression? State this in the table details.

Response) The p-values were added to the manuscript. After logistic regression

analysis, BMI was associated with the composite outcome (odds ratio 0.97, 95%

confidence interval 0.94–1.00, per kg/m2 of BMI, P-value = 0.044). However, BMI was

not known to the RRT nurse at the time of PAR assessment, therefore did not influence

PAR. “The patients exhibited male predominance (60.6%), a median age of 72 (IQR, 61–79)

years, a mean body mass index of 21.4 (IQR, 19.4–24.7) kg/m2, and a median

Charlson comorbidity index of 2 (IQR, 1–4). Patients who died and/or admitted to the

ICU seemed to have lower body mass index than those who did not (median 21.2 vs.

22.0, P-value=0.020). Sex, age, and Charlson comorbidity index did not differ

significantly between the two groups (P-values 0.173, 0.310, and 0.427, respectively).”

CORRECTED

3) Table 1, don’t need to state detailed criteria for rapid response team in the caption

CORRECTED

4) The details of how the RRT work at the hospital is long and extremely detailed. I’m not sure we need all the shift times.

CORRECTED

5) What is a part-time intensivist? Is this a critical care attending physician? Is it a moon-lighter in another specialty? The term is imprecise and I do not know what it means. Also, there was mention of ICU fellows, but in what subspecialty? Critical care medicine?

Response) Part-time intensivist refers to physicians who participate in the RRT only

part-time. They attend to other clinical work such as outpatient clinic when they are offshift.

ICU fellows refer to clinical fellows participating in the management of ICU

patients, and they specialize in pulmonology, nephrology, and emergency medicine.

Such descriptions seemed redundant, therefore were deleted from the manuscript.

AGREE

6) When were clinical variables for the various scores obtained? At bedside or last vital sign check? Same day labs or admission labs?

Response) The clinical variables for the scores were obtained from the latest alerted

vital signs and/or lab findings upon RRT activation, which was on the same day.

“Details of each score calculation are available in the Tables S1–S4 of the supporting

information. Patients’ vital signs used in score calculation were recorded by ward

nurses; these data were immediately sent to the RRT. The RRT nurses and physicians

were not aware of these scores at the time of their visits and assessments, and the

scores did not influence clinical decisions.”

CORRECTED

7) Imprecise language in discussion paragraph 3 (page 12). “the possibility that non-physician healthcare professionals can be helpful in the RRT setting”. Certainly no one would argue that a non-physician healthcare professional is not helpful in an RRT. I don’t understand what the point of the statement is but it could be construed as hurtful to imply the care team is not helpful.

CORRECTED

8) Discussion paragraph 3, page 12: ACGME duty hours is not referenced. I thought that the ACGME went back to allowing 28 hour shifts for interns. There needs to be an accurate depiction of what the policy is with up to date citations.

NEW COMMENT: I disagree with this line of argument. The ACGME 80 work limit has been in effect for a very long time. There was a policy to no 24 hour shifts for interns in 2011 that was revoked in the recent guidelines. There is a lot of data on increased hand offs over time in healthcare, don’t go down this pathway as it is confusing. Cite a large healthcare quality or management study about this issue.

7. PLOS authors have the option to publish the peer review history of their article (what does this mean?). If published, this will include your full peer review and any attached files.

Reviewer #1: Yes: Amr Salah Omar

Reviewer #2: No

Reviewer #3: No

---

## [Author Response · Author response to Decision Letter 1]

7 Oct 2019

Dear editor: 

We would like to thank all the editors and reviewers for providing us with their valuable opinions about our manuscript. We have revised our manuscript according to the comments and recommendations of the reviewers. We have highlighted all changes in the revised manuscript in red font, and a clean copy of the manuscript is also uploaded. Below, we have included an itemized series of responses to the comments of the reviewers.

Reviewer #2: 

Materials and Methods

1. Lines 125-132: The response to my original comment “Please give more detail on the logistic regression methods i.e. how variables were selected, entered, and removed from, the final models” has not been adequately addressed. It remains unclear which variables were considered for inclusion in the logistic regression modelling of the composite outcome, and the rationale behind these considerations.

Response) We appreciate your comment. The combined logistic regression models with PAR and other early warning scores were considered (i.e. PAR + MEWS, PAR + ViEWS, PAR + SEWS, and PAR + CART). Among these, the early warning score with the highest AUROC, ViEWS, was selected to be combined with PAR. We clarified the sentence as follows (Lines 129–132):

“A combined model of PAR with other early warning scores (MEWS, ViEWS, SEWS, and CART) were considered for logistic regression analysis. Among them, ViEWS, which showed the largest AUROC among the calculated scores, was selected, and the AUROC of this model was calculated as well.”

2. Lines 174-179: Regarding the response to my original comment to “Please provide the probabilities associated with the analyses of differences in AUROC’s between the five measures for the ICU admission and mortality outcomes” the authors have added the estimates and CI’s of the individual early warning sign methods which is to be welcomed – but they have not added the probability levels for either of these tests of group differences in AUROC as they have done for the composite outcome dependent variable.

Response) Thank you for sharing the great idea. We have added the p-values for each comparison: PAR against other warning scores. The sentences were amended as follows (Lines 171–182):

“In predicting the composite outcome, PAR provided an AUROC of 0.87 with a 95% confidence interval (CI) of 0.84–0.89, which was significantly higher compared to MEWS (AUROC, 0.66 [95% CI, 0.62–0.70]; P<0.001), ViEWS (AUROC, 0.69 [95% CI, 0.66–0.73]; P<0.001), SEWS (AUROC, 0.67 [95% CI, 0.63–0.70]; P<0.001) and CART (AUROC, 0.63 [95% CI, 0.59–0.66]; P<0.001) (Fig 1). PAR was superior in anticipating ICU admission (AUROC, 0.87 [95% CI, 0.84–0.89]) than MEWS (AUROC, 0.65 [95% CI, 0.61–0.69], P<0.001), ViEWS (AUROC, 0.68 [95% CI 0.64–0.72]; P<0.001), SEWS (AUROC, 0.66 [95% CI 0.62–0.70]; P<0.001), and CART (AUROC, 0.63 [95% CI 0.59–0.67], P<0.001). Although MEWS was superior in predicting mortality (AUROC, 0.79 [95% CI 0.70–0.87]) than CART (AUROC, 0.58 [95% CI 0.49–0.67]; P=0.001), it was not superior than MEWS (AUROC, 0.69 [95% CI 0.61–0.76]; P=0.062), ViEWS (AUROC, 0.70 [95% CI 0.61–0.78]; P=0.101), or SEWS (AUROC, 0.69 [95% CI 0.62–0.77]; P=0.073) (Table 2).”

3. Lines 208-220: Similarly, regarding the response to my original comment “Please quote the probabilities associated with these three analyses” the authors have provided the estimates and CI’s but not the p values.

Response) Thank you again. The p-value was added as follows (Lines 210–214):

“The AUROCs were larger for nurses 1, 2 and 3 with longer experience in the RRT (AUROC, 0.91 95% CI 0.87–0.95], 0.91 [95% CI 0.85–0.96] and 0.90 [95% CI 0.86–0.94] respectively) than for nurses 4 and 5, with shorter experience (AUROC, 0.78 [95% CI 0.72–0.84] and 0.80 [95% CI 0.71–0.90] respectively). AUROC of PAR for nurse 4 was significantly smaller compared to those of nurses 1, 2 and 3 (P=0.003).”

4. Sentence commencing at Line 272: Regarding the response to my original comment “The authors should acknowledge the limitations of their data when discussing the importance of nurse work experience and the accuracy of PAR scores. Specifically, they identified two nurses with less RRT experience whose PAR score accuracy was compared with three more experienced RRT nurses. Such a small sample size (of nurses) cannot provide an adequate test of the impact of experience on PAR accuracy so greater caution in generalising this finding is called for. The least experienced nurses might have had other characteristics which were of equal or greater relevance to the question of their PAR accuracy” the authors have made improvements to the text to qualify this area of their analysis. They should go further by adding at the end of the sentence commencing at Line 272 a further clause such as “...although this would need to be tested in appropriately designed future studies”

Response) Thank you for your careful reading. We have added the phrase as follows (Lines 274–277):

“Although the small number of nurses makes statistical comparison impossible, it seems that sufficient work experience as an RRT nurse may be an important component of a nurse’s ability to correctly predict patient prognosis; this needs to be tested in appropriately designed studies in the future.”

Reviewer #3

1. I think that the authors have de-emphasized the experience factor, however, see comments above. If you truly de-emphasize the splitting of one nurse, your study conclusion changes = PAR+ViEWS is superior to PAR.

Response) We understood your concern, and the authors undertook deep discussion with this issue. With help of a qualified statistician from Medical Research Collaborating Center of Seoul National University Bundang Hospital, we agreed that AUROC is likely to rise with addition of other factors within the same group. Therefore, PAR+ViEWS is highly likely to have larger AUROC in predicting the composite outcome, compared to PAR or ViEWS alone, when compared within the same population. This problem could be solved by dividing the population into two: derivation cohort and validation cohort. Logistic regression model should be derived from the derivation cohort, and the AUROC must be calculated within the validation cohort. 

With all due respect to the reviewer, however, comparison of PAR+ViEWS against PAR was not the main focus of our study. We aimed to validate the performance of PAR, an already existing subjective scoring system, by comparing it against well-known objective scoring systems. Introduction of PAR+ViEWS was to carefully suggest a method to back-up the innate flaws of PAR: its subjective nature. Inexperienced nurses should be supervised, and well-known scoring systems could be utilized for this purpose. 

We agree that de-emphasizing the experience of nurses could mislead the readers. Sentences were amended as follows (Lines 215–219, 271–277, and 306–310):

“To compensate for the smaller AUROC of less-experienced nurses’ PAR, we utilized a logistic regression model of PAR with the warning score of the largest AUROC in our study: ViEWS. In the overall population, the combined model of PAR and ViEWS showed improved AUROC (0.875 [95% CI, 0.849–0.900]) compared to that of PAR alone (0.868 [95% CI 0.843–0.894]); however, this should be interpreted with caution due to the lack of separation of derivation and validation groups.”

“One of the major drawbacks of PAR is its subjective nature. As shown in our study, the reliability of PAR was questionable: PAR of nurses with longer experience in the RRT revealed an AUROC of over 0.9 in predicting mortality and/or ICU admission within the next day, but those with less experience (<6 months to 1 year) showed an AUROC of under 0.8. Although the small number of nurses makes statistical comparison impossible, it seems that sufficient work experience as an RRT nurse may be an important component of a nurse’s ability to correctly predict patient prognosis; this needs to be tested in appropriately designed studies in the future.”

“Conclusions

In conclusion, subjective assessment of the patient by an experienced RRT nurse, represented as PAR, reveals good performance in predicting patient prognosis. Although early warning scores may be useful for identifying at-risk patients, direct examinations by healthcare professionals should be emphasized when RRT is activated.”

2. The PAR+ViEWS is superior to all of these scores?

Response) Thank you for your idea. Various combinations of scores were compared: PAR+MEWS, PAR+ViEWS, PAR+SEWS, and PAR+CART. The AUROC and 95% CI were as follows: 0.875 (0.849–0.900) for PAR+MEWS, 0.875 (0.850–0.900) for PAR+ViEWS, 0.874 (0.848–0.899) for PAR+SEWS, and 0.873 (0.847–0.898) for PAR+CART. The differences of AUROCs between these combinations were not significant (P=0.710). 

3. Not every facility has nurses with a large amount of experience in an RRT team (in fact all of the nurses were quite experienced overall). There is such a small sample size that eliminating one of the nurses because they did not score similar to the others is not possible here. The PAR+ViEWS was superior to the PAR. The risk of n=5 is you have outliers; conversely, your group with a large amount of direct RRT-experienced nurses is probably less like RRT teams around the world – where there could be very inexperienced nurses. Again, this is the risk you run with your trial. The fact that if you take out the inexperienced nurse is a small discussion point, not a major conclusion.

Response) We deeply understand your concern, and agree that addition of ViEWS to PAR is only a small discussion point, rather than a major conclusion. We answered this issue along with the response to comment 1.

4. I would also state this in limitations – that you did not collect direct data on the PAR used for final triage by the physician/and or final decision maker.

Response) We appreciate your comment. The limitation section was amended as follows (Lines 294–298):

“This study has several limitations. First, this study had a single-center design and included only 5 RRT nurses with long working experience, and thus, our results may not be generalizable to other healthcare professionals or less experienced nurses. Inter-observer variability should be considered, and further studies on healthcare professionals with various backgrounds can enforce the strength of PAR.”

5. Were any further outcomes assessed? This should be mentioned in the limitation section, that we do not have this data and ICU admission as an outcome does not account for inappropriate/unneeded ICU admissions. Really, the outcome that is more important is mortality and functional status.

Response) Thank you for the comment. A sentence was added to the limitations section as follows (Lines 303–304):

“Fourth, outcomes other than short-term prognosis, including unnecessary ICU admissions, long-term mortality, and functional status, were not included.”

6. The ACGME 80 work limit has been in effect for a very long time. There was a policy to no 24 hour shifts for interns in 2011 that was revoked in the recent guidelines. There is a lot of data on increased hand offs over time in healthcare, don’t go down this pathway as it is confusing. Cite a large healthcare quality or management study about this issue.

Response) We deeply appreciate your idea. The reference was replaced with a systematic review by West CP and his colleagues, published in Lancet, 2016. The sentence was revised as follows (Lines 248–250):

“To prevent and reduce physician burnout, placing limitations on duty hours can be effective [20], which may lead to an increased need for non-physician healthcare professionals.”

---

## [Decision Letter · Decision Letter 2]

24 Oct 2019

PONE-D-19-17205R2

Performance of Patient Acuity Rating by Rapid Response Team Nurses for Predicting Short-Term Prognosis

PLOS ONE

Dear Yeon Joo Lee,

Thank you for submitting your manuscript to PLOS ONE. After careful consideration, we feel that it has merit but does not fully meet PLOS ONE’s publication criteria as it currently stands. Therefore, we invite you to submit a revised version of the manuscript that addresses the points raised during the review process.

ACADEMIC EDITOR: The reviewers have still raised a number of points which we believe minor modifications are necessary to improve the revised manuscript, taking into account the reviewers' remarks. Please consider and address each of the comments raised by the reviewers before resubmitting the revised manuscript.

We would appreciate receiving your revised manuscript by Dec 08 2019 11:59PM. To enhance the reproducibility of your results, we recommend that if applicable you deposit your laboratory protocols in protocols.io, where a protocol can be assigned its own identifier (DOI) such that it can be cited independently in the future. For instructions see: http://journals.plos.org/plosone/s/submission-guidelines#loc-laboratory-protocols

We look forward to receiving your revised manuscript.

Kind regards,

Wisit Cheungpasitporn, MD, FACP

Academic Editor

PLOS ONE

Reviewers' comments:

Reviewer's Responses to Questions

**Comments to the Author**

1. If the authors have adequately addressed your comments raised in a previous round of review and you feel that this manuscript is now acceptable for publication, you may indicate that here to bypass the “Comments to the Author” section, enter your conflict of interest statement in the “Confidential to Editor” section, and submit your "Accept" recommendation.

Reviewer #2: (No Response)

Reviewer #4: (No Response)

2. Is the manuscript technically sound, and do the data support the conclusions?

Reviewer #2: Partly

Reviewer #4: Yes

3. Has the statistical analysis been performed appropriately and rigorously? 

Reviewer #2: Yes

Reviewer #4: Yes

4. Have the authors made all data underlying the findings in their manuscript fully available?

Reviewer #2: Yes

Reviewer #4: Yes

5. Is the manuscript presented in an intelligible fashion and written in standard English?

Reviewer #2: Yes

Reviewer #4: Yes

6. Review Comments to the Author

Reviewer #2: OVERALL IMPRESSION

The authors have addressed my comments relating to the previous submission which has improved the manuscript. My only reservation is the part of the manuscript which focuses on the impact of the experience of the participating nurses on the predictive power of PAR scores when there is such a small sample size (n=5 nurses). The manuscript would be improved by removal of these results and related sections of the discussion altogether in order to maintain emphasis on the principal finding.

MAJOR COMMENTS

Materials and Methods

Comment 1: Lines 192-132: the authors have added information which adequately clarifies the rational for selection of variables for their logistic regression modelling.

MINOR COMMENTS

Results

Comment 2: Lines 171-182: the authors have added the p-values for these analyses as suggested.

Comment 3: Lines 210-214: the authors have added a p-value for this analysis as suggested.

Discussion

Comment 4: Lines 274-277: the authors have adequately addressed my previous suggestion by the addition of the clause “: this needs to be tested in appropriately designed studies in the future”

Reviewer #4: the authors have addressed the raised issues, no further comment. all comments were appropriate and welcome accept as is

7. PLOS authors have the option to publish the peer review history of their article (what does this mean?). If published, this will include your full peer review and any attached files.

Reviewer #2: No

Reviewer #4: No

---

## [Author Response · Author response to Decision Letter 2]

30 Oct 2019

Dear editor: 

We would like to thank all the editors and reviewers for providing us with their valuable opinions about our manuscript. We have revised our manuscript according to the comments and recommendations of the reviewers. We have highlighted all changes in the revised manuscript with “Track changes” function, and a clean copy of the manuscript is also uploaded. Below, we have included an itemized series of responses to the comments of the reviewers.

Reviewer #2: 

The authors have addressed my comments relating to the previous submission which has improved the manuscript. My only reservation is the part of the manuscript which focuses on the impact of the experience of the participating nurses on the predictive power of PAR scores when there is such a small sample size (n=5 nurses). The manuscript would be improved by removal of these results and related sections of the discussion altogether in order to maintain emphasis on the principal finding.

Response) We appreciate your comment. After careful discussion by the authors, we have decided to remove the contents associated with the impact of the experience of the participating nurses on the predictive power or PAR. The relevant sentences were deleted throughout the manuscript, along with Table 4.

---

## [Editor Report · Decision Letter 3]

31 Oct 2019

Performance of Patient Acuity Rating by Rapid Response Team Nurses for Predicting Short-Term Prognosis

PONE-D-19-17205R3

Dear Dr. Yeon Joo Lee,

We are pleased to inform you that your manuscript has been judged scientifically suitable for publication and will be formally accepted for publication once it complies with all outstanding technical requirements.

With kind regards,

Wisit Cheungpasitporn, MD, FACP, FASN

University of Mississippi Medical Center

Twitter: @wisit661 Email: wcheungpasitporn@gmail.com 

Academic Editor

PLOS ONE

Additional Editor Comments:

I want to commend the authors on their superb efforts to revise the manuscript according to all reviewers’ suggestions. The quality of the manuscript has improved substantially.

Reviewers' comments:

N/A

---

## [Editor Report · Acceptance letter]

6 Nov 2019

PONE-D-19-17205R3 

Performance of Patient Acuity Rating by Rapid Response Team Nurses for Predicting Short-Term Prognosis 

Dear Dr. Lee:

I am pleased to inform you that your manuscript has been deemed suitable for publication in PLOS ONE. Congratulations! Your manuscript is now with our production department. 

With kind regards,

on behalf of

Dr. Wisit Cheungpasitporn 

Academic Editor

PLOS ONE